# Impact of impurities and cryoconite on the optical properties of the Morteratsch glacier (Swiss Alps)

Biagio Di Mauro[1,*], Giovanni Baccolo[2,3], Roberto Garzonio[1], Claudia Giardino[4], Dario Massabò[5,6], Andrea Piazzalunga[7], Micol Rossini[1], Roberto Colombo[1]

[1] Remote Sensing of Environmental Dynamics Laboratory, Earth and Environmental Sciences Department, University of Milano-Bicocca, 20126 Milan, Italy;

[2] Earth and Environmental Sciences Department, University of Milano-Bicocca, 20126 Milan, Italy;

[3] National Institute of Nuclear Physics (INFN), Section of Milano-Bicocca, Milan, Italy;

[4] Institute for Electromagnetic Sensing of the Environment, National Research Council of Italy, Milan, Italy;

[5] Department of Physics, University of Genoa, Genoa, Italy;

[6] National Institute of Nuclear Physics (INFN), Genoa, Italy;

[7] Water & Life Lab SRL, Entratico (BG), Italy

* *Correspondence to*: Biagio Di Mauro (biagio.dimauro@unimib.it)

**Abstract.** The amount of reflected energy by snow and ice plays a fundamental role in their melting processes. Different non-ice materials (carbonaceous particles, mineral dust, microorganisms, algae etc.) can decrease the reflectance of snow and ice promoting the melt. The object of this paper is to assess the capability of field and satellite (EO-1 Hyperion) hyperspectral data to characterize the impact of light-absorbing impurities on the surface reflectance of ice and snow of the Vadret da Morteratsch, a large valley glacier in the Swiss Alps. The spatial distribution of both narrow-band and broad-band indices derived from Hyperion was analyzed in relation to ice and snow impurities. In situ and laboratory reflectance spectra were acquired to characterize the optical properties of ice and cryoconite samples. The concentrations of Elemental Carbon (EC), Organic Carbon (OC) and Levoglucosan were also determined to characterize the impurities found in cryoconite. Multi-wavelength absorbance spectra were measured to compare the optical properties of cryoconite samples and local moraine sediments. In situ reflectance spectra showed that the presence of impurities reduced ice reflectance in visible wavelengths of 80-90%. Satellite data also showed the outcropping of dust during the melting season in the upper parts of the glacier, revealing that seasonal input of atmospheric dust can decrease the reflectance also in the accumulation zone of the glacier. The presence of EC and OC in cryoconite samples suggests a relevant role of carbonaceous and organic material in the darkening of the ablation zone. This darkening effect is added to that caused by fine debris from lateral moraines, assumed to represent a large fraction of cryoconite. Possible input of anthropogenic activity cannot be excluded and further researches are needed to assess the role of human activities in the darkening process of glaciers observed in recent years.

## 1 Introduction

Mountain glaciers represent an important source of fresh water across the planet. These resources are seriously threatened by global climate change (Immerzeel et al., 2010), and a widespread reduction of glacier extension has been observed in recent years (Oerlemans, 2005; Paul et al., 2004). Surface processes that promote glacier melting are driven by both climate (i.e. temperature and precipitation) and changes in albedo. The latter is mainly influenced by the metamorphism of snow in the infrared part of the reflectance spectrum of snow, and by the impurity content (such as dust, soot, ash, algae etc.) in the visible domain (Painter et al., 2007). The studies on the impact of light-absorbing impurities (LAI) on the cryosphere at global (Flanner et al., 2007, 2009), regional (He et al., 2014; Lee et al., 2017; Painter et al., 2010; Sterle et al., 2013) and local scale (Oerlemans et al., 2009) are suggesting that LAI accumulation on snow and ice decreases the albedo with consequences on the radiative and mass balances, both in mountain glaciers and ice sheets (Dumont et al., 2014; Gabbi et al., 2015; Wittmann et al., 2017).

Carbonaceous particles such as Black Carbon (BC), Elemental Carbon (EC) and Organic Carbon (OC) represent an important class of LAI because, in accordance to their chemical structure, they absorb visible light with extreme efficiency (Andreae and Gelencsér, 2006; Hartmann et al., 2013). Their role in the climatic system has been largely acknowledged in the scientific literature (e.g. Bond et al., 2013). Given their capability to absorb visible light, carbonaceous particles are responsible of a positive radiative forcing when deposited on snow and ice (Doherty et al., 2010; Hadley and Kirchstetter, 2012; Meinander et al., 2014). Recently, Painter et al. (2013a) combined ice core data and climate simulations, suggesting that carbonaceous LAI depositions may have played a crucial role in fostering the end of the Little Ice Age in the European Alps.

Recent studies showed that alpine glaciers are undergoing a darkening process, and this was ascribed to the impact of regional/global warming and to the deposition and accumulation of LAI on the glacial surface (Azzoni et al., 2016; Gautam et al., 2013; Ming et al., 2012, 2015; Qu et al., 2014). Impurities such as BC are commonly present in the atmosphere in Alpine regions (Lavanchy et al., 1999; Ménégoz et al., 2014; Nair et al., 2013; Sandrini et al., 2014). Their sources can be both natural (e.g. biomass burning, wildfires) and anthropogenic (e.g. fossil fuel and bio-fuel combustion). Instead, mineral dust (MD) originates from natural sources including areas surrounding the glaciers or distant arid regions (e.g. deserts). The determination of the composition of non-ice material is fundamental to attribute the provenance of these particles and to assess the actual role of anthropogenic and natural activity in decreasing the albedo of glaciers and accelerating the melt.

When impurities are deposited on snow and ice, they can aggregate on the surface of the glacier forming a characteristic sediment defined as "cryoconite" (Nordenskiöld, 1883). Cryoconite is constituted of dust and organic matter so it decreases the albedo of the ice and promotes its melting (Takeuchi et al., 2001). Cryoconite can accumulate in typical holes in the ablation areas of mountain and polar glaciers (i.e. "cryoconite holes"), or can be dispersed on their surface (Stibal et al., 2012). Its geochemical and microbiological composition has been studied in polar and non-polar glaciers (Aoki et al., 2014; Bøggild et al., 2010; Hodson et al., 2007; Nagatsuka et al., 2014; Takeuchi et al., 2014; Wientjes et al., 2011) to evaluate its provenance and to determine the role of microorganisms (i.e. extremophile) in sustaining life in such harsh environments. Nevertheless, a lot of uncertainty still exists regarding its formation and evolution. Moreover, most of the literature focused on Arctic, Antarctic and Asian glaciers, while few information is available for the European Alps (Edwards et al., 2013; Franzetti et al., 2016) although their high sensitivity to environmental and climatic changes (Beniston, 2005). The Alps rise on the top of the Po river plain, one of the most industrialized and densely populated areas of Europe. As a consequence, the presence of anthropogenic activities is expected to strongly influence the

geochemical and radiative characteristics of cryoconite (Baccolo et al., 2017; Cook et al., 2015; Hodson, 2014; Tieber et al., 2009).

Within this study, we demonstrate the potential of field and satellite hyperspectral data to characterize the spatial distribution of impurities and cryoconite on the surface of the Vadret da Morteratsch glacier (Swiss Alps) and to evaluate their effect on snow and ice reflectance. Specific objectives consist in: i) the analysis of the variability of the Morteratsch glacier surface reflectance combining hyperspectral satellite data and field spectroscopy data; ii) the exploitation of different narrow-band and broad-band indices to determine the impact of impurities and cryoconite on the glacier; and iii) the characterization of the optical and chemical properties of cryoconite in the study area.

## 2  Data and Methods

### 2.1  Study area

Vadret da Morteratsch (46°24′34″N, 9°55′54″E, hereafter referred as Morteratsch) is a glacier located in the Bernina range in the European Alps (Fig. 1) (Oerlemans and Klok, 2002). Morteratsch is a large valley glacier with an altitude that spans from 2030 to 3976 m a.s.l., with an area of 15.81 km$^2$. It is a typical north facing Alpine glacier experiencing negative mass balances in the last century (Nemec et al., 2009; Oerlemans et al., 2009; WGMS, 2013). The glacier apparatus is constituted by the Morteratsch glacier and of its tributary, the Pers glacier. A long record of measurements is showing that Morteratsch glacier front retreated of 2.65 km since 1878  (VAW/ETHZ & EKK/SCNAT, 2016). Modeling simulations based on emission scenarios predicted an almost total deglaciation of Morteratsch at the end of the century (Zekollari et al., 2014).

<Figure 1>

### 2.2  Field campaign

A field campaign was conducted on July 30[th] 2015 on the ablation zone of the Morteratsch glacier (see Fig. 1). Field spectral measurements were collected to characterize the spectral reflectance of the glacier surface characterized by different physical conditions and impurity content. Four main classes were identified by visual inspection: clean ice, dirty ice, melt pond and cryoconite. A minimum of three different points were measured for each class. Spectral reflectance was measured with an Analytical Spectral Devices (ASD) field spectrometer (Hand Held), which collects reflected radiance from 325 to 1075 nm with a spectral sampling interval of 1 nm and a spectral resolution (full width at half maximum, FWHM) of 3.5 nm at the band centered at 700 nm. The hemispherical conical reflectance factor (HCRF) $\rho(\lambda)$ was calculated by normalizing the reflected radiance $L_\uparrow(\lambda)$ with the incident radiance $L_\downarrow(\lambda)$ measured from a calibrated Spectralon© panel. Each spectra was the average value resulting from 15 scans, all corrected for the instrument dark current. Three replicates were acquired at each point and a minimum of three different points were measured for each of the four classes. HCRF spectra were all obtained between 12.00 and 13.00 (local time) under clear-sky conditions in order to minimize the uncertainty related to variations in the radiation environment (e.g., changes in the direct/diffuse ratio) during the field measurements. A bare optical fiber with a field of view of 25° was used to collect data from nadir with respect to the surface.  The fiber optic was held by a fiber holder equipped with a level. The fiber holder was always kept at a distance of 80 cm from the ground corresponding to a footprint diameter of 35cm. As a measure of the ASD reflectance measurements uncertainty, we calculated the coefficient of variation averaged on the VIS-NIR wavelengths. In our study, the coefficient of variation spans from 1 to 10%.

Samples of ice, cryoconite and debris from the lateral moraine were collected (three samples for each class) in the areas measured with the field spectrometer. Ice sampling was conducted crushing the surface ice, cryoconite samples were picked using a small spoon, lateral moraine samples were collected near the glacier terminus. All samples were stored in sterilized Corning tubes (50 mL). They were kept frozen and taken back to the University Campus (Milano-Bicocca), where they were stored at -30 °C. A gravimetric method was then used to estimate the overall load of impurities from samples of clean and dirty ice. For the gravimetric determination of cryoconite concentration we estimated an error equal to 4% by repeating 5 times the measurement

### 2.3    Cryoconite optical and chemical characterization

Samples of cryoconite collected from the glacier surface were stored in frozen conditions until the preparation of the samples for the analyses. After melting, they were decanted for several hours to separate solid and liquid fractions. The liquid part was removed, and solid cryoconite was successively dried at 60°C for 4 hours. Dried samples were re-suspended and filtered on quartz fiber filters (Pall, 2500QAO-UP, 47 mm diameter) to obtain a thin homogeneous layer suitable for light absorption measurements as well as thermo-optical analysis (TOT). Even if we kept a relatively low drying temperature to avoid sample modifications, we cannot definitely exclude that some compounds with very high volatility may have been removed with the drying process. However, organics do not volatilize at temperatures lower than 100°C, thus, if we lost some organics with the drying process, they should be compounds that are in the gas phase at ambient temperature, and that usually constitute a minimum fraction with respect to the total OC. Moreover, the filters were weighed before and after the extraction so that the mass concentration is simply given by difference. The weighing was performed after a period of 48 hours of conditioning in controlled conditions room (T ~ 20 ± 1 °C; RH ~ 50 ± 5%), with an analytical balance (Sartorius model MC5, precision of 1 µg); the electrostatic effects are removed exposing the filter before weighing to a de-ionizer. After weighing, the loaded filters were first subjected to a multi-wavelength optical analysis performed by the Multi-Wavelength Absorbance Analyzer (MWAA, a full description can be found in Massabò et al., 2013, 2015). With the MWAA, the absorbance (Abs) has been retrieved for each sample at five wavelengths (λ = 375, 407, 532, 635, 850 nm), measuring each filter at 64 different points, each ~ 1 mm$^2$ wide. The Mass Absorption Cross-section (MAC, in m$^2$/g) was estimated by normalizing for the mass of the samples. For MWAA measurements, the uncertainty is about 10%, and it is given by the squared sum of the uncertainty related to the surface variability of the sample (~5%) and the uncertainty of the optical measurement (calculated as 3 times the variability of the blanks, and equal to 8%).

For the Elemental Carbon (EC) and Organic Carbon (OC) determination a TOT instrument (Sunset Lab Inc.) was used applying the NIOSH 5040 protocol (Birch and Cary, 1996). According to the instrument manual, an uncertainty of 8-10% is associated to the retrieved EC and OC concentrations. The original solutions have been subjected to a chemical determination of the Levoglucosan (1,6-anhydro-b-D-glucopyranose) concentration by High Performance Anion Exchange Chromatography coupled with Pulsed Amperometric Detection (for more details, see Piazzalunga et al., 2010). Levoglucosan is known to be a marker of biomass burning (Kehrwald et al., 2012), and was used here to evaluate the source of the carbon contained in cryoconite samples.

Hyperspectral reflectance spectra were acquired also in laboratory on dried cryoconite and moraine sediment samples. Each sample was lighted with a halogen stable light source (1000 W, LOT Quantum Design). The lamp produces a uniform collimated output beam (with 50 mm diameter) that provides a homogenous stable illumination of the sampled

area. Reflectance was measured with an ASD field spectrometer (Full Spec PRO) that, differently from the one used in the field, operates from 350 to 2500 nm with a FWHM of 5–10 nm and a spectral sampling of 1 nm.

## 2.4 Satellite data

Remotely sensed data used in this study were collected with the hyperspectral Hyperion sensor on board on the NASA Earth Observing One (EO-1) satellite mission, launched in November 2000 (Middleton et al., 2013). Hyperion sensor features a swath of 7.7 km and a spatial resolution of 30 meters. It collects spectral radiance from 400 to 2400 nm with a spectral resolution of 10 nm (242 bands). The signal-to-noise ratio (SNR) of Hyperion data varies from 150:1 (for 400-1000 nm) to 60:1 (for 1000-2000 nm). Hyperion reflectance retrievals have been validated several times with independent measurements, also with airborne sensor such as AVIRIS (Kruse et al., 2003).

The Hyperion image was acquired on 7th August 2015 (close to the field campaign) with a foot print covering the sites visited in the field campaign and comprising the whole Morteratsch, Pers, Fellaria glaciers and also other minor glaciers (Fig. 1). The look angle of Hyperion was 23° during the acquisition, and the solar zenith angle (SZA) was equal to 50°. The image was atmospherically corrected using the 6SV code (Kotchenova et al., 2006; Kotchenova and Vermote, 2007; Vermote et al., 1997). The code was parameterized with Aerosol Optical Depth (AOD) retrieved by interpolated measurements gathered from nearby Aerosol Robotic Network (AERONET) stations (i.e. Davos and Ispra), a continental aerosol model and a midlatitude climatic profile.

The goodness of the Hyperion atmospheric correction was evaluated with concurrent Landsat 8 Operational Land Imager (OLI) data acquired few minutes before Hyperion on August 7th. The OLI surface reflectance Climate Data Record (CDR) product, already corrected for the influence of the atmosphere (Vermote et al., 2016) was compared with atmospherically corrected Hyperion data. Hyperion scene was classified with respect to land cover using a Support Vector Machine (SVM) algorithm (Wu et al., 2004). SVM is a supervised classification method derived from statistical learning theory. The algorithm separates the classes with a decision surface (i.e. optimal hyperplane) that maximizes the margin between the classes. The full Hyperion spectrum was used as input for the spectral discrimination of the classes. SVM has already been successfully applied to complex and noisy data such as the Hyperion ones (Petropoulos et al., 2012). We manually selected the training set for eight pre-defined land cover classes: snow, bare ice, debris cover, lakes, rocks, and sparse vegetation; clouds and shadow were also included. The goodness of the classification was evaluated by considering 100 randomized points (weighted for each class) and building a confusion matrix to estimate the producer's and user's accuracies for each class, as well as the global accuracy of the SVM classification. For the two main classes of interest (snow and bare ice), the ratio between training and test set pixels was ~ 10%. For the Hyperion pixels classified as snow and ice, narrow-band and broad-band indices were calculated as described below.

From Hyperion surface reflectance data, the Snow Darkening Index (SDI) (Di Mauro et al., 2015), was calculated as a normalized difference between red and green bands (Equation 1).

$$SDI = (\rho_{red} - \rho_{green})/(\rho_{red} + \rho_{green}) \qquad \text{(Eq. 1)}$$

The SDI is nonlinearly correlated with MD concentration in snow. Positive SDI values are associated with the presence of MD on snow, conversely negative values represent clean snow. Red and green bands were averaged considering the following wavelengths: 640, 650, 660, 671 nm in the first case, and 548, 559, 569,579 nm in the second case.

The Impurity index (I$_{imp}$) (Dumont et al. 2014) was derived from Hyperion spectra, as the ratio between logarithms of reflectance in the green and near infrared regions (Equation 2)

$$I_{imp} = \log(\rho_{green}) / \log(\rho_{NIR}) \qquad \text{(Eq. 2)}$$

I$_{imp}$ was developed to monitor the impact of LAI on the Greenland Ice Sheet from MODIS satellite data. From Hyperion spectra we averaged reflectance at 548, 559 and 569 for the green and 844, 854 and 864 for the near infrared regions, respectively.

Finally, Hyperion data were used to estimate the glacier albedo ($\alpha_{VIS}$) by integrating the Hyperion spectral reflectance in the visible wavelengths ($\Delta\lambda = 701 - 447$ nm) (Equation 3).

$$\alpha_{VIS} = (\sum_{i=447}^{701} \rho(\lambda_i)(\lambda_{i+1} - \lambda_i) / \Delta\lambda \qquad \text{(Eq. 3)}$$

In this study, we did not characterize the anisotropy of snow on the glacier, so we were not able to perform a proper estimate of hemispherical albedo (see Naegeli et al. 2015), but we used $\alpha_{VIS}$ computed as the numerical integral of HCRF in visible wavelengths (0.4 - 0.7 µm) (Liang, 2001), as an approximation of snow albedo. This calculation was performed assuming a Lambertian behavior of snow and ice reflectance.

The narrow-band (SDI, I$_{imp}$) and broad-band ($\alpha_{VIS}$) indices computed from the Hyperion-derived reflectance were also calculated from ASD-derived spectra. These indices calculated from the ASD reflectance spectra were then compared to the concentration of impurities measured in clean and dirty ice.

### 2.5 Radiative transfer modelling

In order to assess the sensitivity of the SDI, I$_{imp}$ and $\alpha_{VIS}$ to snow grain size (SGS), and BC and MD concentrations, we ran a set of simulations using the Snow, Ice, and Aerosol Radiation (SNICAR) model (Flanner et al., 2007). The model allows to simulate the snow hemispherical albedo spectra between 300 and 5000nm with a resolution of 10 nm. The main variables included in the model are: snow grain size (µm), snow density (Kg/m$^3$), snowpack thickness (m), surface spectral distribution, solar zenith angle (degrees), MD and BC concentration (respectively in ppm and ppb). We simulated snow reflectance varying the SGS from 100 to 600µm, the (uncoated) BC concentration from 0 to 1200ppb and the MD concentration from 0 to 300ppm (diameter 5.0–10.0 µm). SNICAR simulations were run with a direct incident radiation (SZA = 50°) in order to match the Hyperion acquisition. Then we calculated the three indices and represented them as a function of MD/BC concentrations and SGS in a contour matrix plot.

## 3 Results

### 3.1 Spectral characterization of glacier surface at different scales

#### 3.1.1 Field data

Figure 2 shows the spectral behavior of the four classes (clean ice, dirty ice, melt pond, and wet cryoconite) measured in field and the two classes (dry cryoconite and moraine sediment) measured in laboratory. Mean and standard deviation are obtained by averaging spectra of the same class. Clean ice shows an average reflectance in the visible and near infrared wavelengths which is higher with respect to the other classes (Fig. 2a). Dirty ice shows very low reflectance values of circa 0.3 in the visible wavelengths, and lower than 0.2 in the near infrared. Melt pond shows an enhanced absorption at

1000 nm, and wet cryoconite shows the lowest average reflectance values (i.e. <0.07) across all the investigated spectral range (Figure 2b). Dry cryoconite shows overall higher reflectance than the wet cryoconite measured in field; in particular, the reflectance of dry cryoconite increases with the wavelengths. The spectral reflectance of moraine sediments is higher than wet and dry cryoconite, with a strong increase for wavelengths shorter than 560 nm, and an almost constant behavior for larger wavelengths. Clean ice and dirty ice reflectance is almost flat, especially in visible wavelengths, whereas melt pond and wet cryoconite show a maximum at circa 560 nm.

<Figure 2>

The presence of impurities in glacier ice strongly alters its optical properties, reducing its reflectance. Results from the linear ordinary least square (OLS) regressions between SDI, $I_{imp}$, albedo and concentration of impurities measured in clean ice and dirty ice show that $\alpha_{VIS}$ is negatively correlated with the impurity concentration ($R^2 = 0.53$), whereas SDI and $I_{imp}$ are positively correlated ($R^2 = 0.32$ and $R^2 = 0.54$ respectively). Among nonlinear OLS, a power law function maximizes the goodness of fit ($R^2 = 0.9$) between the $I_{imp}$ and impurity concentrations (Fig. 3).

<Figure 3>

### 3.1.2 Hyperion data

While field spectroscopy enables to characterize with high precision the heterogeneous surface of the ablation zone, remote sensing data allow to investigate the optical properties of the glacier as a whole. A comparison of field and satellite data is shown in Fig. 4, where ASD reflectance of dirty ice is compared with the average spectra of bare ice class from Hyperion data (RMSE =0.03). Unfortunately, we were unable to identify on the Hyperion image a pure region with clean ice in the ablation zone to compare with ASD spectra. Field and satellite spectra are highly comparable in the visible and near infrared wavelengths. The only relevant discrepancy concerns short wavelengths: before 500 nanometers Hyperion spectra show a decrease in reflectance, where ASD spectra remain almost flat (Fig. 4).

<Figure 4>

Figure 5 shows the result of the SVM classification applied to the whole Hyperion scene. SVM algorithm properly separates different surfaces in the Bernina range. Table 1 presents the confusion matrix obtained from the validation scheme. Global accuracy of the classification is 78%. The user's accuracy for the classes snow and bare ice classes is 90% and 70%, while the producer's accuracy is 85.7% and 73.7%. Rocks and debris cover classes are sometimes misclassified, their user accuracy is 60% and 70% respectively. The class with the lower accuracy (40%) is lakes.

<Table 1>

<Figure 5>

Examples of spectra extracted from Hyperion data for classes snow, bare ice, and debris cover for the Morteratsch and Pers glaciers are shown in Fig. 6a. The accumulation zone of the Morteratsch shows an overall higher reflectance in the visible and near infrared wavelengths, with enhanced absorption features at 1030 nm and 1250 nm. Beyond 1400 nm almost all the radiation is absorbed by snow. The accumulation zone of the Pers glacier shows a lower reflectance in the visible and near infrared wavelength, with stronger absorption before 600 nm. Ice reflectance from both Morteratsch and Pers glaciers is lower than snow reflectance; in particular, the absorption at 1030 nm is more pronounced and shifted to lower wavelengths. The reflectance spectrum of debris cover shows different features and an overall higher reflectance

in the short wave infrared. Figure 6b shows a comparison between the atmospherically corrected reflectance from Hyperion and Landsat 8 OLI (mean and standard deviation) (RMSE = 0.015). The areas selected are the same of Figure 6a for the Morteratsch glacier. Despite some noise in the Hyperion spectra, data from the two satellites are in good agreement, both for the three OLI bands (B2, B3, B4) in the visible domain and for the OLI band (B5) in the infrared domain.

<Figure 6>

Figure 7 shows the $\alpha_{VIS}$, $I_{imp}$ and SDI maps obtained from atmospherically corrected Hyperion data. In all glaciers, the accumulation zone shows higher $\alpha_{VIS}$ values (0.9-1) than the ablation zone ($\alpha_{VIS}$~0.3), where bare ice is exposed (see also the normalized frequency distribution in Fig. 7e-g). The inner part of the Morteratsch ablation zone shows slightly higher values ($\alpha_{VIS}$~0.4) than those close to the debris covered areas. The highest $\alpha_{VIS}$ values (0.9-1) are observed in the large accumulation zone of the Fellaria glacier, on the south face of the massif. Regarding $I_{imp}$ and SDI maps, higher values are associated with stronger presence of impurities and mineral dust respectively. In all glaciers, the accumulation zone generally shows lower $I_{imp}$ than the ablation zone. SDI was developed to track changes in mineral dust content in snow and it allows to evidence patterns of snow impurity in the accumulation zone, while its interpretation over ice in the ablation zone is still uncertain. The lower SDI values are observed in the large accumulation zone of the Fellaria glacier, where also the highest $\alpha_{VIS}$ values are estimated. Figure 7d shows a RGB combination of the three indices, where the red channel is the SDI, the green is the $\alpha_{VIS}$, and the blue is the $I_{imp}$. This representation emphasizes the presence of a large area of the glacier interested by high values of SDI and $I_{imp}$, characterized by magenta hues. Conversely, the upper parts of glaciers are dominated by yellow-green hues, indicating the strong influence of the $\alpha_{VIS}$.

<Figure 7>

### 3.1.3    Sensitivity of narrow- and broad-band indices to SGS and LAI concentrations

In Figure 8, we present the contour plots obtained from the SNICAR simulations. Plots refer to the sensitivity of narrow- and broad-band indices to MD variations (upper panels) and to BC variations (lower panels). $I_{imp}$ is insensitive to SGS for both MD and BC variations. For low concentrations of BC/MD, also SDI is almost insensitive to SGS, but for high concentrations, a nonlinearity emerges. Avis results the most sensitive index to SGS. $I_{imp}$ and $\alpha_{VIS}$ are similarly affected by variations in MD and BC, while SDI is more sensitive to MD than BC.

<Figure 8>

### 3.2    Organic and elemental carbon in cryoconite

Table 2 shows the concentration of Elemental Carbon (EC), Organic Carbon (OC), Total Carbon (TC, calculated as the sum of EC and OC) and Levoglucosan in the cryoconite samples. Different samples show similar values of EC and OC concentrations, respectively ranging from 0.3 to 0.4% for EC, and from 4.2 to 5.4% for OC (Figure 9). Traces of Levoglucosan were found only in cryoconite samples CR5 and CR6.

< Table 2 >

< Figure 9 >

### 3.3 Comparison of the optical properties of cryoconite and moraine sediment

The comparison between the MWWA of a cryoconite (CR4) and a moraine sediment (MS1) sample is presented as an example in Figure 10. The MWWA measures the absorbance (Abs) of samples at five different wavelengths. This parameter quantifies the fraction of light absorbed by the sample and it is defined as Abs = $\tau$ (1-$\omega$), where $\tau$ is the total optical depth and $\omega$ the aerosol-filter layer single scattering albedo (for further information see Petzold and Schönlinner, 2004). The Mass Absorption Cross-section (MAC, in $m^2/g$) was estimated by normalizing for the concentration of the samples. The spectral dependence of the absorbance shows that the particles contained in both the cryoconite samples and in the lateral moraine absorb more at shorter wavelengths with respect to longer ones. In fact, both CR4 and MS1 samples show a decreasing trend in absorbance (Fig. 10a and 10b). This spectral characteristic is also present in the MAC plots (Fig. 10c and 10d). MAC data are shown here for comparison with other studies in the atmospheric sciences, where this variable is fundamental for determining the radiative properties of aerosol particles. In general, MS1 data show lower absorbance and MAC than CR4 data, but this is also valid with respect to the other cryoconite samples (data not shown).

< Figure 10>

### 4 Discussion

Results here presented confirm the effect of impurities and cryoconite in reducing the spectral reflectance of snow and ice in the European Alps (Naegeli et al., 2015). The combined use of ground and satellite hyperspectral observations and the exploitation of narrow-band and broad-band indices provides original data to evaluate the effect of impurities and cryoconite in the Morteratsch glacier. Furthermore, the laboratory determination of the composition of non-ice material suggests an important role of carbonaceous material in the darkening of the Morteratsch ablation zone. This effect is superimposed on the one caused by fine debris from the lateral moraine.

From field spectroscopy, we were able to characterize different glacier components in the ablation zone only, while satellite data allowed to have an overview on the reflectance spatial variability at catchment scale. We tried to select flat areas for the reflectance measurements. However, the surface of the glacier was quite rugged, so possible uncertainties related to the forward scattering of snow may be present in the data. Reflectance higher than 1 in the visible wavelengths is often found in the literature (Painter and Dozier, 2004; Schaepman-Strub et al., 2006), and can be related to forward scattering, glacier surface slope, sensor tilt, inappropriate bidirectional reflectance distribution function (BRDF) correction etc. Over the ablation zone of the Morteratsch glacier, the optical properties of ice are largely variable probably because of the patterns of melting, refreezing and surface runoff that shape the local glacier morphology. The presence of impurities and cryoconite causes a dramatic reduction of the spectral reflectance of ice. In the visible wavelengths, the ice reflectance decreases from circa 1 to 0.3 due to impurity presence and to less than 0.1 due to cryoconite presence (see Fig. 2). The decrease in reflected energy has an important impact on the radiative balance of a retreating glacier such as the Morteratsch (Klok et al., 2004; Oerlemans et al., 2009). We showed that the concentration of impurities and $I_{imp}$ are nonlinearly correlated ($R^2$= 0.9), with concentrations spanning from 1.3 to 9740 ppm (see Fig. 3), and this is in agreement with modeling simulations presented in Dumont et al. (2014). Reflectance spectra of cryoconite measured on the glacier (see Fig. 2b) show very low reflectance value in the visible and near infrared wavelengths. In particular, a maximum reflectance value of 0.06 was found at 550 nm and then a decreasing trend for larger wavelengths. Moraine sediments show an overall higher reflectance with respect to cryoconite samples. Conversely, dry cryoconite samples measured in laboratory show an almost doubled spectral reflectance in the visible and infrared wavelengths. This is due to the fact that the presence of water decreases cryoconite relative refractive index and, as a consequence, wet cryoconite samples are

darker than dry ones (Lekner and Dorf, 1988; Twomey et al., 1986).This process has a big impact on the radiative balance of a glacier, since the presence of melt water and cryoconite material on its surface can add a significant input of absorbed radiation. However, it should be noted that wet cryoconite reflectance is expected to vary as a function of the optical properties of the ice beneath and its wetness, thus the effect of cryoconite presence on glacier albedo is not easily predictable. Previous hyperspectral measurements of cryoconite (Takeuchi, 2002, 2009; Tedesco et al., 2013) showed similar optical properties. In particular, our reflectance spectra are in agreements with those from Tedesco et al. (2009) that measured cryoconite samples collected in Greenland and Antarctica. This suggests that the study of the optical properties of Alpine cryoconite could serve as a reference also for studying their impact on ice sheets.

In this study, we evaluated the reliability of the Hyperion reflectance through a comparison with Landsat and ASD reflectance, and we obtained respectively an RMSE of 0.015 and 0.03. The comparison between field and Hyperion reflectance spectra shows that ASD field spectra of the class dirty ice are comparable with those measured from Hyperion sensor. Hyperion spectra show a decrease of reflectance before 500 nm, this could be due to the presence of contaminated (non-pure) pixels of snow and ice, as previously observed by Negi et al. (2013). Otherwise, it could be related to the presence of meltwater increasing the absorption of solar radiation during the melting season, as observed from airborne hyperspectral reflectance data in other glaciers of the European Alps (Naegeli et al., 2015). Spectra of bare ice exhibit low reflectance, showing an enhanced absorption around 1030 nm, slightly shifted to lower wavelengths (Dumont et al., 2017; Green et al., 2002).

The classification of Hyperion data using the SVM algorithm provides satisfying results for the classes snow and bare ice (user's accuracies of 90 and 70%) confirming the successful use of Hyperion for snow and ice monitoring (Bindschadler and Choi, 2003; Casey et al., 2012; Doggett et al., 2006; Negi et al., 2013; Zhao et al., 2013). For other land cover classes (e.g. rocks and debris cover), a lower user's accuracy was obtained (60-70%), probably due to both the coarse spatial resolution (30m) of Hyperion for mapping these classes, and the spectral similarity between the two classes. Hyperion data provide important information regarding the optical properties of snow and ice in relation to the impact of light-absorbing impurities and grain size. In particular, $\alpha_{VIS}$, $I_{imp}$ and SDI maps show different spatial patterns on the glaciers of the Bernina range, revealing that each index bears different information. From the results of the SNICAR simulations presented in Section 3.1.3 we can assess that $I_{imp}$ is a solid indicator of LAIs concentration. SDI instead is more related to the radiative impact of LAIs on snow, since it is also influenced by the increase of SGS. However, during summer season characterized by wet snow with large SGS, SDI is almost insensitive to changes in SGS, while its sensitivity to medium/low MD concentration is maximum. Furthermore, since SDI is a broad band RGB index, it can be easily estimated from digital RGB cameras both fixed (e.g. Webcam) and mounted on Unmanned Aerial Vehicles. Although SDI and $I_{imp}$ share a common band (at 550-580nm), they emphasize different aspects of the impact of LAIs on snow and ice reflectance. $I_{imp}$ is similarly affected by variations in MD/BC, while SDI is more sensitive to MD variations, in particular for large SGS. In comparing SDI and $I_{imp}$ maps, it should be noted also that both indices are varying with the sun geometry, which is varying on the glacier due to local topography. For example, Dumont et al. (2014) computed $I_{imp}$ from MODIS diffuse albedo to analyse LAIs distribution over the Greenland Ice Sheet.

In agreement with previous satellite investigations on alpine glaciers, low $\alpha_{VIS}$ values are estimated in the Morteratsch ablation zone (Brun et al., 2015; Dumont et al., 2011; Fugazza et al., 2016; Klok et al., 2004; Naegeli et al., 2015, 2017). This decrease in albedo can be explained by both an increase of LAI content and/or variations of the snow/ice grain properties. However, the interpretation of the effects of such external and internal snow characteristics on $\alpha_{vis}$ is not straightforward. For example, Liou et al. (2014) showed that snow grain shape and impurity snow mixing structures can

significantly influence the effects of LAIs on snow albedo. Furthermore, snow grain packing also plays a critical role in affecting albedo of both clean and dirty snow (He et al., 2017). High SDI and $I_{imp}$ values are found across the ELA (located approximately at 3000m, following Zekollari et al., 2014), where Saharan dust layers probably associated to heavy depositions occurred during the hydrologic year 2014-2015 (Di Mauro et al., 2015; Varga et al., 2014) appear on the

glacier surface. The presence of outcropping dust across and above the ELA of an alpine glacier during the summer season further decreases its reflectance and enhances the melt at higher altitude, where new snow and firn are directly exposed to solar radiation. To date, this particular process has been poorly studied, but further research is needed to explore this impact on Alpine mountain ranges, also integrating ice core data and mass balance modeling (Gabbi et al., 2015). SDI was applied to describe the impact of mineral dust on snow radiative properties, exploiting its wavelength-dependent

properties. In this paper, we observe that on the ablation zone multiple processes are involved in the darkening of ice, but SDI is unable to fully capture all of them. In fact, during the summer season mineral fine debris, organic matter and other impurities accumulate on the bare ice strongly reducing its albedo at all wavelengths. On the other hand, $I_{imp}$ shows high values also in the ablation zone, demonstrating to be more suitable than SDI in capturing bare ice darkening. In Fig. 7d, a RGB composition of $\alpha_{VIS}$, SDI and $I_{imp}$ clearly shows that in the median and lower part of the Morteratsch glacier the

impact of impurities is stronger, whereas in the uppermost part of the glacial apparatus, green-yellow hues highlight the presence of clean snow, fallen in the last hydrological season.

The spatial resolution of the optical satellites used in this study (i.e. 30 m) hampers the discrimination of the impact of different impurities in alpine glaciers. New opportunities are now offered by the new ESA Sentinel 2 mission featuring a higher spatial resolution in visible and near infrared wavelengths (Drusch et al., 2012). With the decommission of the

EO-1 mission (February 22, 2017), there will be no hyperspectral sensors orbiting the Earth. Hyperspectral data are very useful for monitoring the optical properties of snow and ice (Painter et al., 2016), and future satellite missions such as EnMAP (Environmental Mapping and Analysis Program) (Stuffler et al., 2007) and PRISMA (Hyperspectral Precursor and Application Mission) (Labate et al., 2009), will provide new important hyperspectral data for cryosphere monitoring both in alpine and polar regions.

A previous work on the Morteratsch glacier (Oerlemans et al., 2009) showed that the ablation zone is undergoing a darkening process. The authors attributed the albedo decrease to the accumulation of dust from the moraines surrounding the glacier. In this study, we show how spectral reflectance is distributed over this glacier and in particular over the ablation zone. The presence of cryoconite and impurities over the bare ice in alpine glaciers plays an important role in albedo decrease and melting enhancement (Cook et al., 2015; Takeuchi et al., 2001). The spectral reflectance of ice

gathered from field measurements showed a decrease from 0.9 to 0.1 in visible wavelengths due to the presence of surface non-ice material. Furthermore, the spectral reflectance of moraine sediments showed overall higher values than both wet and dry cryoconite samples. These results suggest considering the impact of organic and inorganic material in albedo decreasing for glacier mass balance modeling.

Regarding the characterization of cryoconite, the chemical and thermo-optical analyses allow us to produce a first

overview on the composition and the light-absorbing properties of the material contained in cryoconite. The concentrations of OC (5% of the total weight) found in cryoconite are comparable with independent measurements on other glaciers (Stibal et al., 2008; Takeuchi et al., 2001). We found no concentration of EC presented in the literature on cryoconite. The presence of EC (as a proxy of BC) in the cryoconite suggests a possible influence of anthropogenic activities (e.g. fossil fuel combustion) in the formation and evolution of cryoconite. Furthermore, carbonaceous particles

are stronger absorber in visible and near-infrared wavelengths than mineral particles, and they can heavily enhance the radiation absorbed by bare ice during summer season (Li et al., 2017). In Fig. 10, we show the absorbance and MAC of cryoconite and moraine sediments. A considerable difference was found between the two samples, in particular cryoconite samples show absorbance values two times higher than those of the moraine. This suggests that the input of moraine sediment alone could not explain the darkening observed on the glacier surface, as previously proposed (Oerlemans et al., 2009). The interaction between mineral material and microbiological activity, with the consequent accumulation of organic matter in cryoconite, further increases the absorption of solar radiation, promoting the melt. Also MAC values reflect this behavior, since they are calculated normalizing the absorbance with the dry weight of the samples. The presence of Levoglucosan in two cryoconite samples suggests a possible input also from biomass burning aerosols, which may represent an important source of carbon in alpine ranges.

The interaction dynamic between carbonaceous particles and microorganisms living in the cryoconite is still largely unknown (Cook et al., 2015), and the lack of EC and BC measurements in cryoconite in the scientific literature did not allowed us to perform quantitative comparisons with other glaciers. We can assess that the source of carbon can be classified on the basis of the optical properties of particles (Massabò et al., 2015), but new methodologies have to be developed to achieve this comparison for cryoconite samples. The main sources of carbonaceous particles in the atmosphere at mid-latitude are the combustion of fossil fuels and bio-fuels, and biomass burning. The source partitioning of BC and EC is still missing for alpine glaciers, and further research is needed to assess the impact of natural and anthropogenic carbon emissions on the albedo of glaciers in different mountain ranges and on the margin of ice sheets (Musilova et al., 2016; Tedesco et al., 2016).

## 5    Conclusions

In this paper, we show how non-ice materials influence the optical properties of a valley glacier (Morteratsch) in the European Alps. Results from field campaigns and satellite hyperspectral data show that impurities and cryoconite decrease ice spectral reflectance in visible wavelengths from 0.9 to 0.1 and 0.06 respectively, adding a consistent input of energy to the glacier radiative balance. Hyperion satellite data show low $\alpha_{VIS}$ values in the ablation zone of the glacier, and SDI and $I_{imp}$ revealed the outcropping of dust across and above the ELA. The presence of impurities in the accumulation zone could further impact the seasonal mass balance of a retreating glacier. In situ spectra of bare ice and cryoconite show that the light absorption is enhanced when the sediment is wet, which is a common situation during the ablation season in alpine glaciers. Chemical and optical analyses of cryoconite show that this sediment contains both organic and elemental carbon, and that it strongly absorbs radiation in visible and infrared wavelengths. The presence of traces of Levoglucosan suggests a possible influence of biomass burning.

Beside climatic drivers, glacier darkening due to multiple processes can promote further glacier shrinkage, and therefore a considerable loss of fresh water reservoir in the European Alps. Further analyses are needed to assess the possible impact of anthropogenic activities in glacier darkening processes. Field observations represent an important tool to validate satellite retrieval of surface reflectance, and the characterization of non-ice material that decreases the reflectance of ice is fundamental to evaluate their provenance. The impact of organic and inorganic material on snow and ice optical properties is receiving much attention from the scientific community, and the inversion of newly developed radiative transfer models (Cook et al., 2017; Libois et al., 2013) will allow to estimate the concentration of different impurities from future multi and hyperspectral remote sensing observations.

**Data availability:**

Data used in this paper will be made available upon request to the first author (biagio.dimauro@unimib.it)

**Author contributions:**

BDM conceived the idea of the paper, organized the field campaign and wrote the manuscript with discussions and contributions from all the other authors. BDM, GB and RG collected the samples and acquired spectral reflectance on the Morteratsch glacier. MR and CG helped in data interpretation. DM and AP performed the chemical and optical analyses of cryoconite and helped in their interpretation. RC and MR supervised the research.

**Acknowledgements**

We acknowledge L. Ong, E. M. Middleton and P. Campbell (NASA-GSFC) for scheduling the EO-1 satellite data acquisition. The samples from Morteratsch were stored at the EuroCold facility of the Earth and Environmental Science Department of the University of Milano-Bicocca. We thank the PIs of AERONET stations of Davos (N. Kouremeti) and Ispra (G. Zibordi). We also thank the Editor and the two anonymous Reviewers for the comments on a previous version of the manuscript.

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

**Tables:**

| | snow | bare ice | debris cover | clouds | shadows | lakes | rocks | sparse vegetation | tot | Producer's Accuracy (%) |
|---|---|---|---|---|---|---|---|---|---|---|
| snow | **18** | 3 | | | | | | | 21 | 85.7 |
| bare ice | 1 | **14** | | | | 4 | | | 19 | 73.7 |
| debris cover | | 2 | **6** | | | | 1 | | 9 | 66.6 |
| clouds | 1 | 1 | | **10** | | | 2 | | 14 | 71.4 |
| shadows | | | | | **10** | 2 | | | 12 | 83.3 |
| lakes | | | | | | **4** | | | 4 | 100 |
| rocks | | | 4 | | | | **7** | 1 | 12 | 58.33 |
| sparse vegetation | | | | | | | | **9** | 9 | 100 |
| tot | 20 | 20 | 10 | 10 | 10 | 10 | 10 | 10 | | |
| User's Accuracy (%) | 90 | 70 | 60 | 100 | 100 | 40 | 70 | 90 | | Global Accuracy 78% |

**Table 1 Confusion matrix obtained from the validation scheme of the Support Vector Machine (SVM) classification of Hyperion data. Rows represent the true classes, and columns represent the predicted classes. User and producer accuracies are reported for each class. Global accuracy is also displayed.**

| | Elemental Carbon (EC) | | Organic Carbon (OC) | | Total Carbon (TC) | Levoglucosan |
|---|---|---|---|---|---|---|
| | ppm | % tot weight | ppm | % tot weight | ppm | mL/L |
| CR3 | 4101 | 0.4% | 42529 | 4% | 46629 | / |
| CR4 | 3089 | 0.3% | 53729 | 5.4% | 56818 | / |
| CR5 | 3952 | 0.4% | 48591 | 4.9% | 52544 | 7.982 |
| CR6 | 3355 | 0.3% | 50745 | 5.1% | 54100 | 8.423 |

**Table 2 Summary of Elemental Carbon (EC), Organic Carbon (OC), Total Carbon (TC) and Levoglucosan concentrations in cryoconite samples (CR3-6)**

**Figures:**

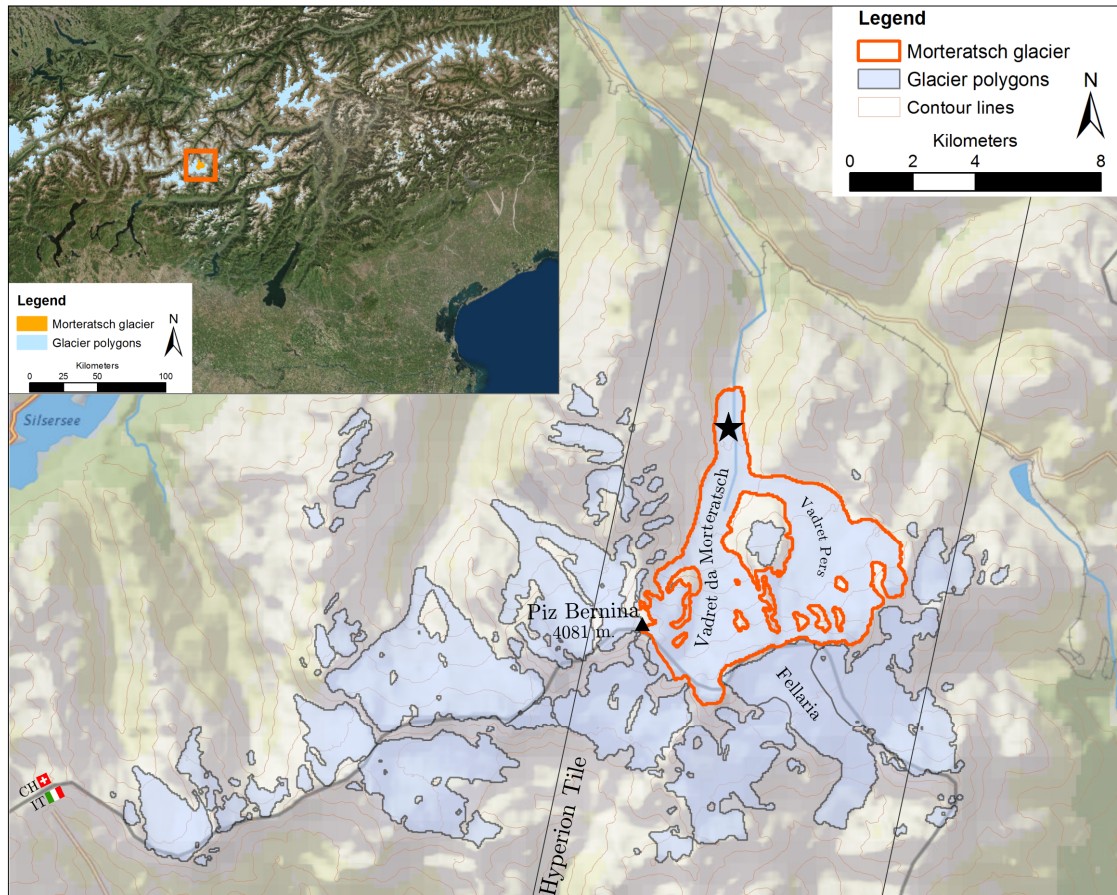

**Figure 1. Location of the Morteratsch glacier in the Bernina range. Glacier polygons are extracted from the Randolph Glaciers Inventory v.5** (Pfeffer et al., 2014)**. Oblique lines represent the extension of the Hyperion scene acquired on August 2015. The black star represents the area where field spectroscopy measurements and sampling were conducted. Also Vedret Pers and Fellaria glacier are displayed and considered in the satellite data analysis**

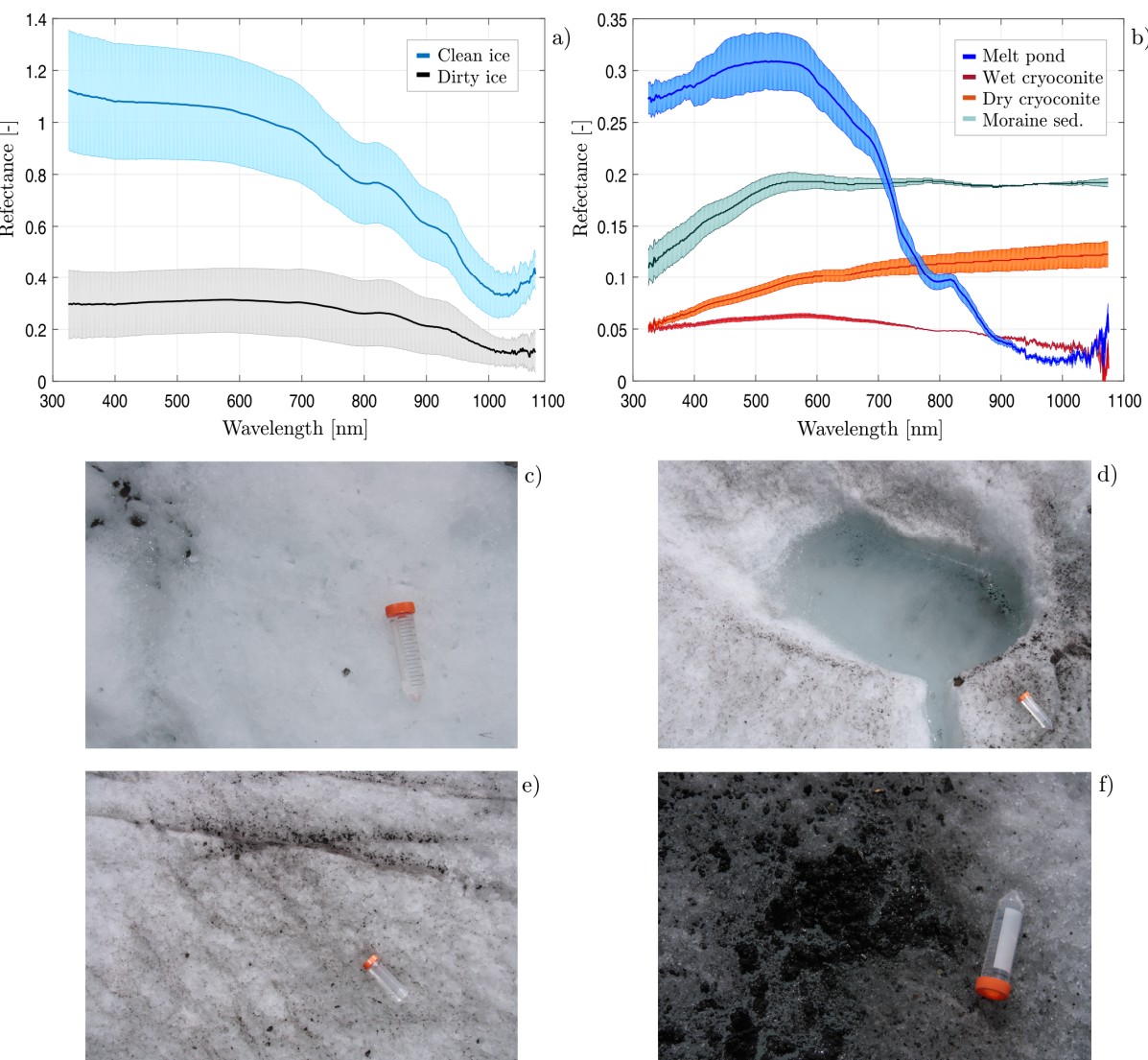

**Figure 2 Mean and standard deviation of the reflectance spectra acquired on the glacier ablation zone with the Field Spec ASD; a): clean and dirty ice (picture in Fig. 2c and 2e) reflectance spectra; b): melt pond and wet cryoconite (picture in Fig. 2d and 2f), dry cryoconite and moraine sediment reflectance spectra (the last two classes were measured in laboratory). Mean and standard deviation were computed on three samples for each target.**

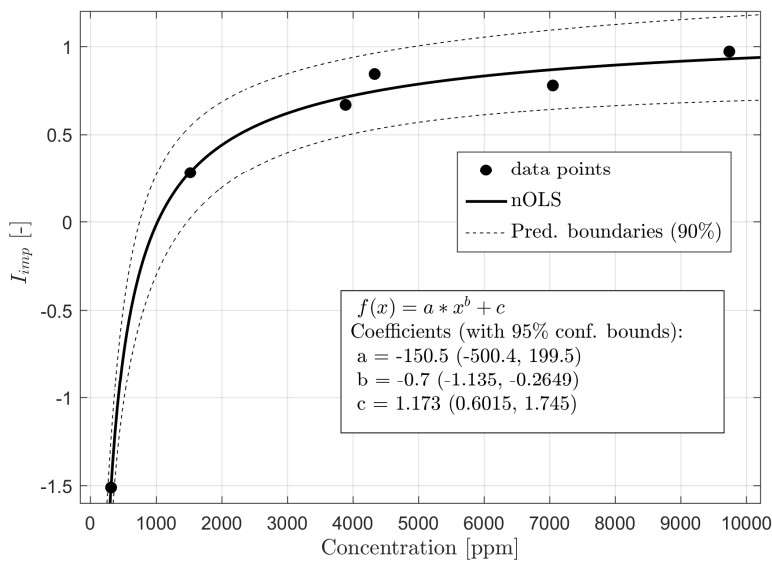

**Figure 3 Nonlinear Ordinary Least Square (nOLS) regression ($R^2 = 0.9$) between the impurity index ($I_{imp}$) and the concentration of impurities found in samples of clean and dirty ice. Model description is displayed in the box.**

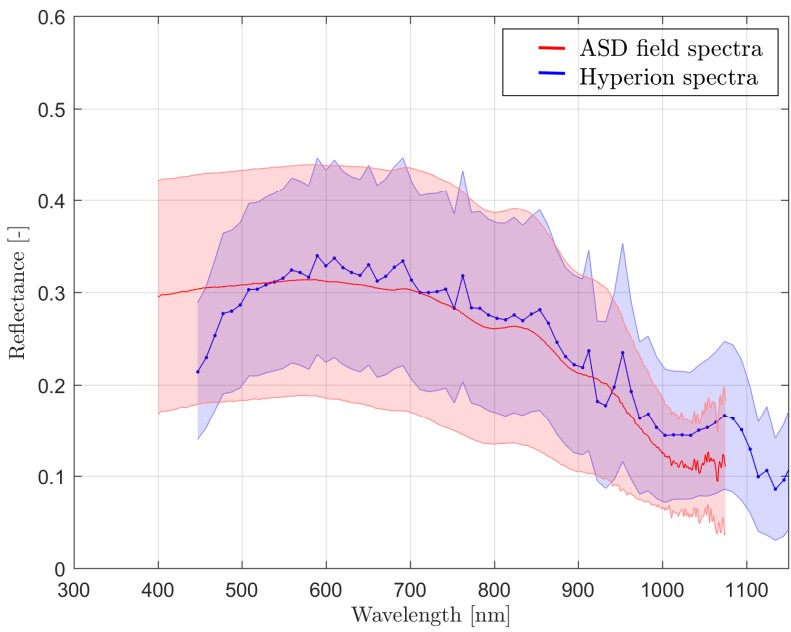

5 **Figure 4 Comparison between ASD field reflectance spectra (red line) for the class dirty ice, and Hyperion reflectance spectra (blue line) for the class bare ice.**

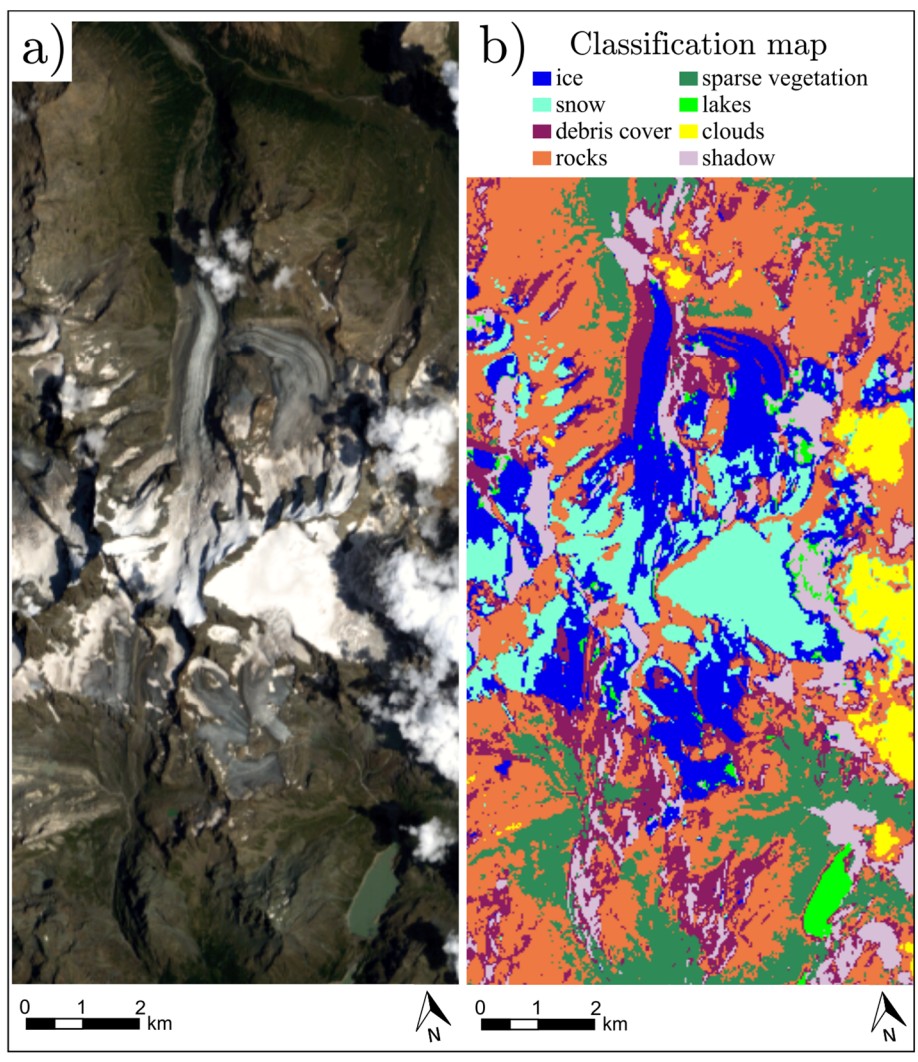

**Figure 5 (a): true color representation of the Hyperion tile over the Bernina range. (b): classification map obtained using the Support Vector Machine (SVM) algorithm.**

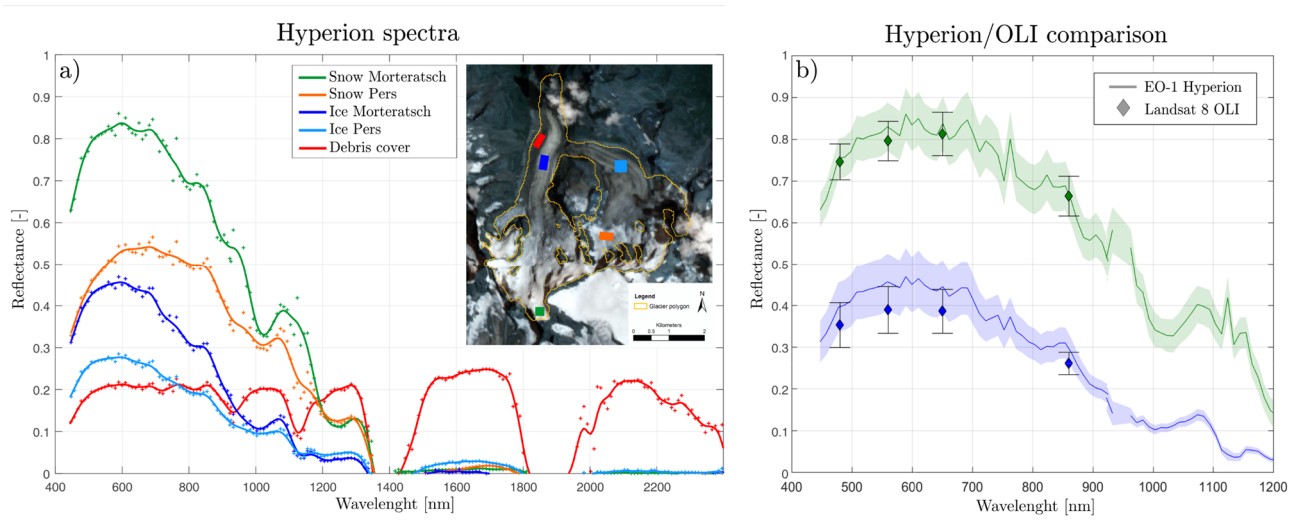

5    **Figure 6 (a): Reflectance spectra of the Morteratsch glacier extracted from the Hyperion scene. Solid lines are smoothed reflectance with a Savitsky-Golay filter in order to remove the noise. (b): Comparison between Landsat 8 OLI bands (diamonds) and EO-1 Hyperion spectra (solid lines) for the ablation (blue) and accumulation (green) zone. Error bars and shaded bands represent the standard deviation of the selected area.**

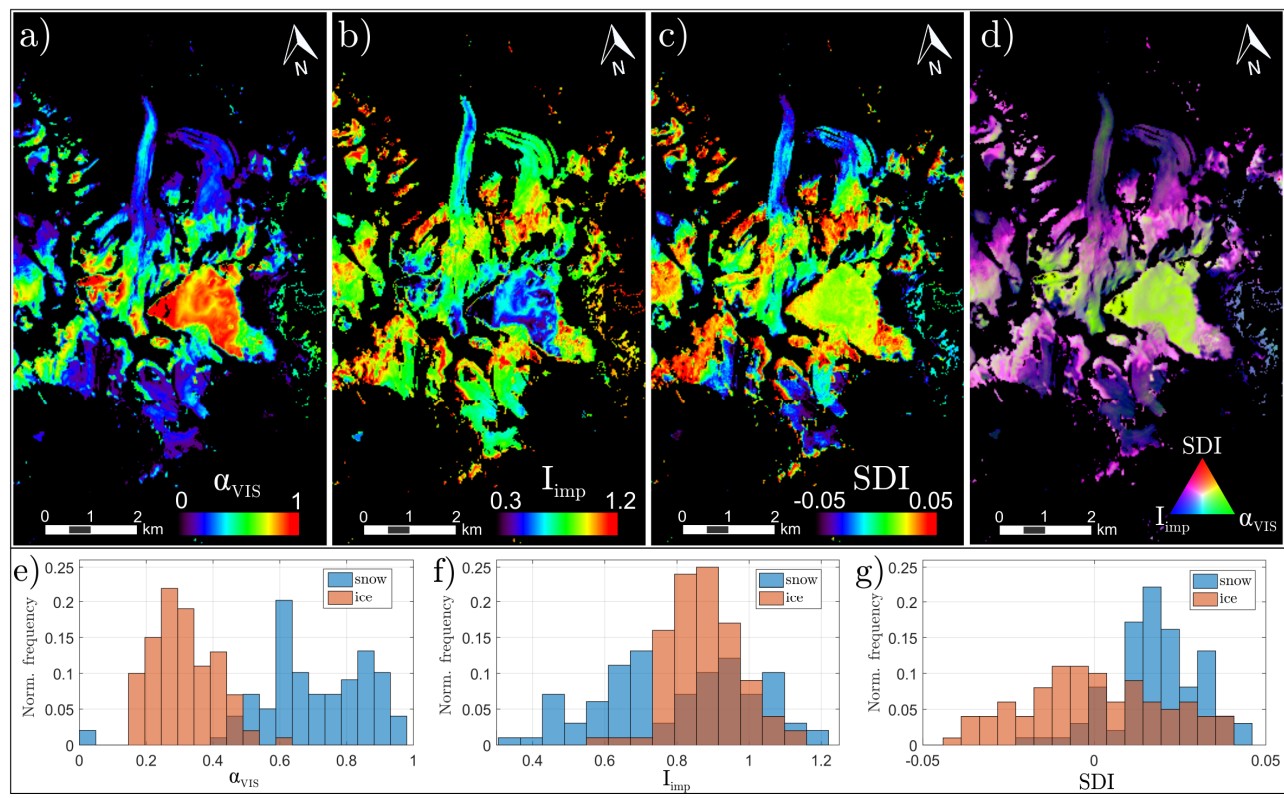

**Figure 7 (a,b,c) Maps of α_VIS, I_imp and SDI obtained from Hyperion reflectance for the classes snow and bare ice in the Bernina range. (d) RGB composition created by combining albedo, I_imp and SDI respectively in the R, G and B channels. (e, f, g) normalized frequency of α_VIS, I_imp and SDI for the classes snow (red bars) and ice (blue bars).**

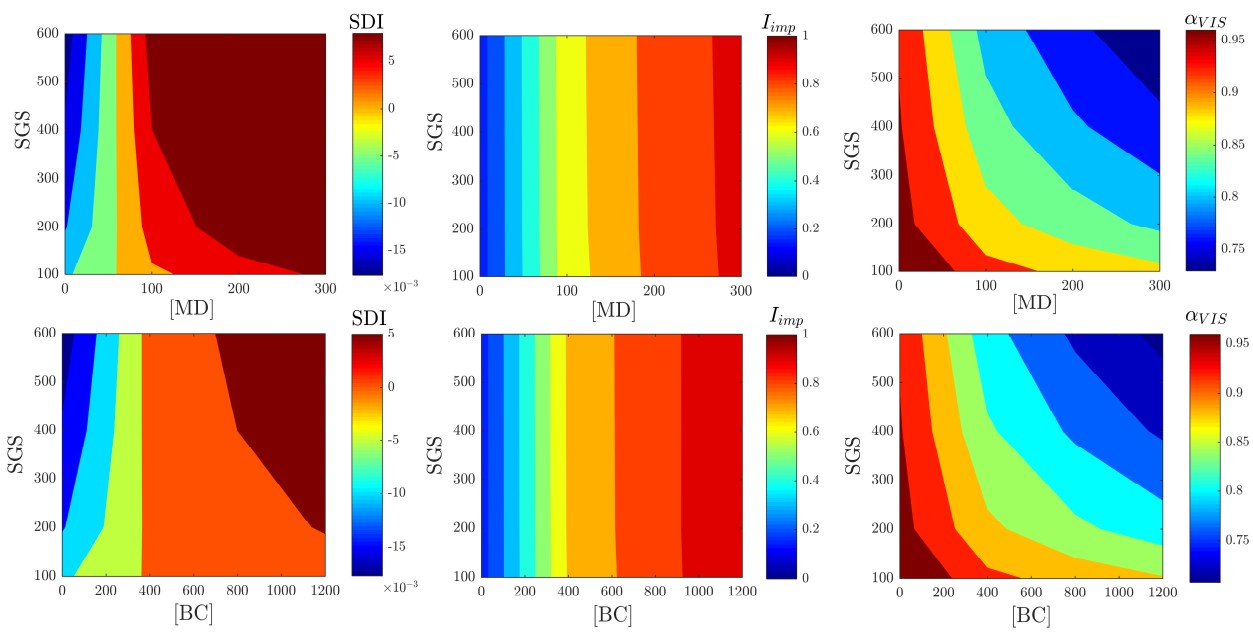

**Figure 8 Comparison between SDI, I_imp and α_VIS obtained from SNICAR simulations. The indexes are represented as contour plots as a function of the concentration of Mineral Dust (MD, upper panels) and Black Carbon (BC, lower panels). MD concentrations are in ppm, BC concentrations are in ppb.**

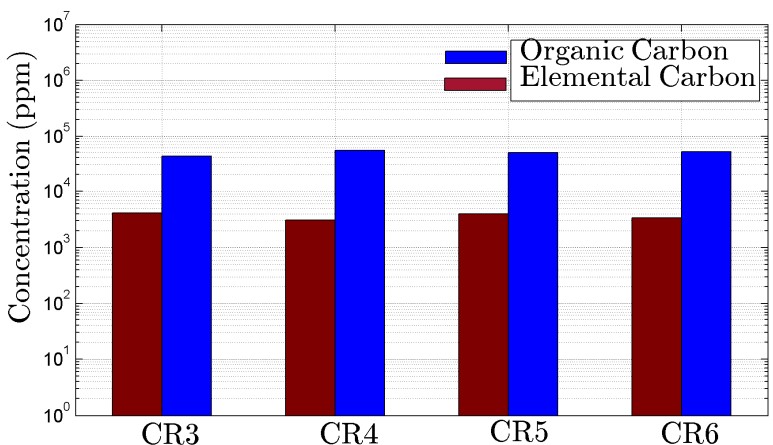

**Figure 9 Concentration of Organic Carbon (blue bars) and Elemental Carbon (red bars) for different cryoconite samples (CR3-CR6)**

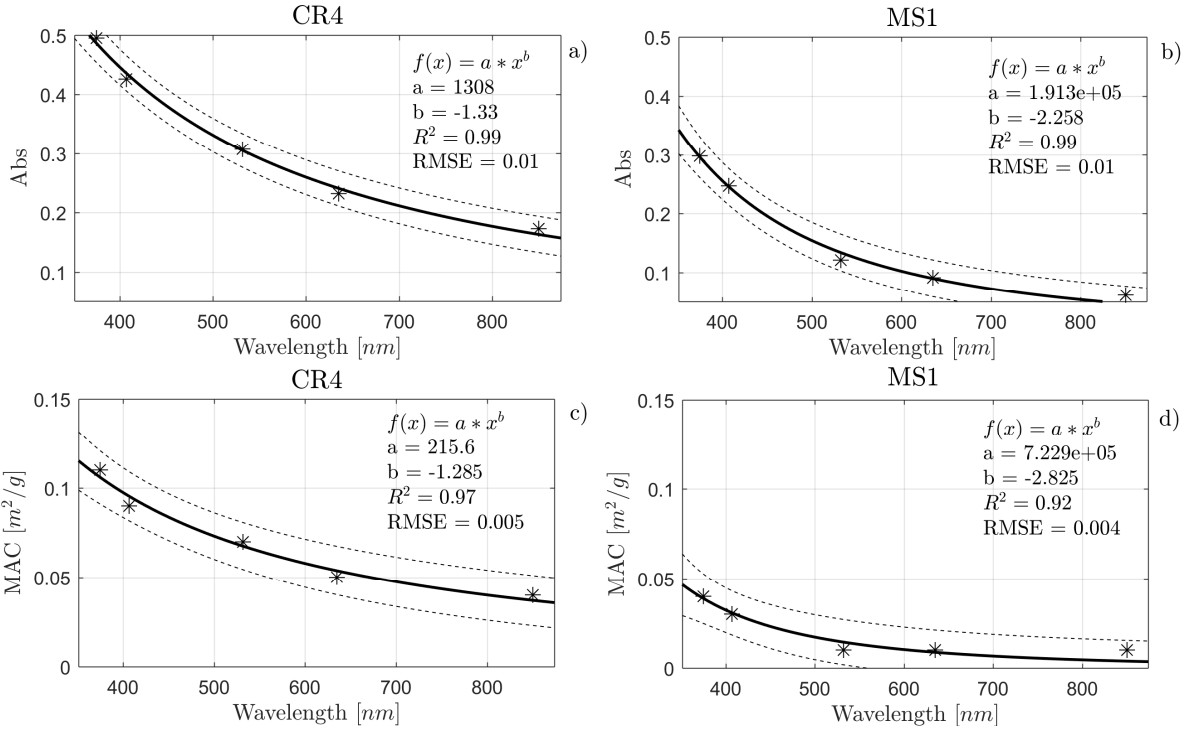

**Figure 10 (a-b): absorbance (Abs) of a cryoconite (CR4) and a moraine sediment (MS1) sample. (c-d): Mass Absorption Cross-section (MAC) of the same samples. Solid lines represent power law fits, dashed lines are the prediction boundaries at 90%.**