# Peer review of "Impact of impurities and cryoconite on the optical properties of the Morteratsch glacier (Swiss Alps)"

_The Cryosphere, 2017_

## Referee Comment (RC1) · Anonymous Referee #1 · 30 May 2017

The authors used field campaigns and satellite hyperspectral data to investigate the effects of impurities and cryoconite on spectral reflectance of snow and ice. They also conducted lab measurements of optical properties of ice and cryoconite samples, which is related to the impurity content in snow/ice. This study provides a good method to characterize the impact of impurities on snow/ice spectral reflectance by combining field, lab, and satellite measurements, which have an important implication for future study. Before this manuscript can be considered for publication, I have a few comments for the authors to address.

General comments:

1. In the methodology section, the authors provided a detailed description of laboratory, field, and satellite measurement processes, which, however, lacks necessary discussions on the uncertainties associated with these measurements. I suggest that the authors add some discussions on this aspect.

2. The authors used the characteristic spectral reflectance of clean and dirty snow/ice to infer the effect of impurities in snow/ice. However, both external (e.g., impurity content) and internal (e.g., snow/ice grain properties) factors can affect the spectral reflectance. For example, Liou et al. (2014) showed that snow grain shape and impurity-snow mixing structures can significantly influence the effects of impurities on snow albedo. He et al. (2017) further found that snow grain packing also plays a critical role in affecting albedos of both clean and dirty snow. Therefore, such internal factors could potentially affect the interpretation of the spectral observations presented by the authors. It would be informative and useful if the authors could include these recent studies and add some discussions on this issue.

Reference:

He, C., Y. Takano, and K. N. Liou (2017), Close packing effects on clean and dirty snow albedo and associated climatic implications, Geophys. Res. Lett., 44, doi:10.1002/2017GL072916.

Liou, K. N., Y. Takano, C. He, P. Yang, L. R. Leung, Y. Gu, and W. L. Lee (2014): Stochastic parameterization for light absorption by internally mixed BC/dust in snow grains for application to climate models, J. Geophys. Res.-Atmos., 119, doi:10.1002/2014JD021665

Specific comments:

1. Page 3, Line 29: "The spectra were all obtained around midday under clear-sky conditions." Are there any specific reasons or advantages to obtain spectra in midday with clear sky?

2. Page 4, Line 4: "solid cryoconite was successively dried at 60°C for 4 hours". Would

this drying process remove some of the organics with relatively high volatility?

3. Page 5, Lines 8–12: What is the percentage of total data points used for SVM training and testing set, respectively?

4. Page 5, Line 30: The indices (narrow- and broad-band) were compared to the impurity concentrations. The indices derived from the Hyperion spectra have a spatial resolution of 30 meters, while the impurity concentration is from point measurement. This is not an apple-to-apple comparison, which may introduce uncertainty. Could the authors discuss this issue?

5. Page 6, Lines 28–29: "The only relevant discrepancy ... where ASD spectra re-main almost flat." Are there any possible explanations for this discrepancy at short wavelengths?

6. Page 7, Section 3.2: The authors only presented the concentration of EC and OC in this section, which seems to lack of the descriptions on the linkage between EC/OC concentration and reduced reflectance. This may confuse the readers. It would be helpful if the authors could explicitly articulate the relationship between EC/OC content and albedo reduction, after the description of EC/OC concentrations in this section.

7. Page 2, Lines 5–10: for the authors' information, a recent study (Lee et al., 2016) combined satellite measurements and model simulations to show the reduced snow albedo caused by impurities over the southern Tibetan Plateau, which could be cited here as a useful reference source.

Reference:

Lee, W. L., K. N. Liou, C. He, S.-C. Liang, Z. Liu, Q. Yue (2016): Impact of absorbing aerosol deposition on snow albedo reduction over the southern Tibetan Plateau Based on Satellite Observations, Theor. Appl. Climatol., 1-10, 10.1007/s00704-016-1860-4.

---

## Author Comment (AC1) · 16 Jun 2017

Authors responses are in *italic,* Reviewer's comments are in **bold.**

**The authors used field campaigns and satellite hyperspectral data to investigate the effects of impurities and cryoconite on spectral reflectance of snow and ice. They also conducted lab measurements of optical properties of ice and cryoconite samples, which is related to the impurity content in snow/ice. This study provides a good method to characterize the impact of impurities on snow/ice spectral reflectance by combining field, lab, and satellite measurements, which have an important implication for future study. Before this manuscript can be considered for publication, I have a few comments for the authors to address.**

*Dear Reviewer #1,*

*Thank you for the positive evaluation of the manuscript. We have carefully considered each of the Reviewer's comments and suggestions. The Reviewer will find below the responses to the specific comments.*

**General comments:**

**1. In the methodology section, the authors provided a detailed description of laboratory, field, and satellite measurement processes, which, however, lacks necessary discussions on the uncertainties associated with these measurements. I suggest that the authors add some discussions on this aspect.**

*Thank you for this comment. All our measurements feature different uncertainties. Hereafter, the uncertainty related to our measurements is discussed. We will add this information in the Methodology section of the manuscript.*

*For the gravimetric determination of cryoconite concentration we estimated an error equal to 4% by repeating 5 times the measurement. Regarding EC/OC determination, uncertainty values are estimated from the software of the instrument, and are generally equal to 8-10%; referencing from the manual: "Calculated errors are based on long-term historical data for replicates and instrumental blanks. Over the course of hundreds of replicate runs, the relative standard deviation is typically 5%". For MWAA measurements, the uncertainty is about 10%, and it is given by the squared sum of the uncertainty related to the surface variability of the sample (~5%) and the uncertainty of the optical measurement (calculated as 3 times the variability of the blanks, and equal to 8%).*

*Regarding ASD data, measurement error is reduced by internally averaging 15 scans for each acquired spectrum, all corrected for the instrument dark current (See Pag. 3 lines 27-28). Field spectra were acquired at midday in clear sky conditions, so the uncertainty related to variability of the incoming radiation should be in principle minimized. Furthermore, three replicas were collected for each sample. We calculated the mean and standard deviation from radiance values and we obtained a coefficient of variation (averaged on VIS-NIR wavelengths) that spans from 1 to 10%. Regarding satellite hyperspectral data, we directly compared Hyperion reflectance with those measured from Landsat and ASD, and we obtained satisfying results. Hyperion reflectance retrievals have been validated several times with independent measurements, also with airborne sensor such as AVIRIS (see for example Kruse et al. 2003). The signal-to-noise ratio (SNR) of Hyperion data varies from 150:1 (for 400-1000 nm) to 60:1 (for 1000-2000 nm); other possible source of uncertainty may come from the atmospheric correction.*

*Ref:*

*Kruse, F. A., Boardman, J. W., & Huntington, J. F. (2003). Comparison of airborne hyperspectral data and EO-1 Hyperion for mineral mapping. IEEE Transactions on Geoscience and Remote Sensing, 41(6), 1388-1400.*

**2. The authors used the characteristic spectral reflectance of clean and dirty snow/ice to infer the effect of impurities in snow/ice. However, both external (e.g., impurity content) and internal (e.g., snow/ice grain properties) factors can affect the spectral reflectance. For example, Liou et al. (2014) showed that snow grain shape and impurity snow mixing structures can significantly influence the effects of impurities on snow albedo. He et al. (2017) further found that snow grain packing also plays a critical role in affecting albedos of both clean and dirty snow. Therefore, such internal factors could potentially affect the interpretation of the spectral observations presented by the authors. It would be informative and useful if the authors could include these recent studies and add some discussions on this issue.**

**Reference:**

**He, C., Y. Takano, and K. N. Liou (2017), Close packing effects on clean and dirty snow albedo and associated climatic implications, *Geophys. Res. Lett.*, 44, doi:10.1002/2017GL072916.**

**Liou, K. N., Y. Takano, C. He, P. Yang, L. R. Leung, Y. Gu, and W. L. Lee (2014): Stochastic parameterization for light absorption by internally mixed BC/dust in snow grains for application to climate models, *J. Geophys. Res.-Atmos.*, 119, doi:10.1002/2014JD021665**

We acknowledge that both internal and external factors impact snow and ice spectral reflectance. In particular, internal factors may play an important role in decreasing the reflectance of ice and snow during long and hot summers at mid-latitudes. We will add a brief discussion on these aspects and we will include the suggested papers in the bibliography.

**Specific comments:**

**1. Page 3, Line 29: "The spectra were all obtained around midday under clear-sky conditions." Are there any specific reasons or advantages to obtain spectra in midday with clear sky?**

Atmospheric disturbance is an important source of error in field spectroscopy. Incoming radiation in field environment is strongly anisotropic and it is a combination of direct/diffuse sunlight scattered from the sky and adjacent objects. This scattering events produce wavelength-dependent effects. A consequence of this is that HCRF measured in the field is subject to uncertainty introduced by the irradiation environment, and are therefore not only related to properties of the surface (see Milton et al. 2009). For this reason, we measured field spectra in clear sky conditions in order to minimize the uncertainty related to the direct/diffuse ratio during the field measurements. The choice of collecting measurements around midday is motivated by the fact that snow and ice have a strong directional effect (see for example Painter & Dozier 2004), and measuring with the Sun at nadir should minimize this source of error.

Ref:

Milton, E. J., Schaepman, M. E., Anderson, K., Kneubühler, M., & Fox, N. (2009). Progress in field spectroscopy. Remote Sensing of Environment, 113, S92-S109.

Painter, T. H., & Dozier, J. (2004). Measurements of the hemispherical-directional reflectance of snow at fine spectral and angular resolution. Journal of Geophysical Research: Atmospheres, 109(D18).

**2. Page 4, Line 4: "solid cryoconite was successively dried at 60∘C for 4 hours". Would this drying process remove some of the organics with relatively high volatility?**

*We cannot exclude that some compounds with very high volatility may be removed with the drying process, we kept this relatively low temperature in order to avoid sample modifications. In any case, using the Sunset system we observed that organics do not volatilize at temperatures lower than 100°C. With this in mind, if we lost some organics with the drying process, they should be compounds that are in the gas phase at ambient temperature, and that usually constitute a minimum fraction with respect to the total OC.*

**3. Page 5, Lines 8−12: What is the percentage of total data points used for SVM training and testing set, respectively?**

*For the two main classes of interest (snow and bare ice), the ratio between training and test set pixel is ~ 10%. We will include this information in the paper.*

**4. Page 5, Line 30: The indices (narrow- and broad-band) were compared to the impurity concentrations. The indices derived from the Hyperion spectra have a spatial resolution of 30 meters, while the impurity concentration is from point measurement. This is not an apple-to-apple comparison, which may introduce uncertainty. Could the authors discuss this issue?**

*For this comparison, indices were calculated from the ASD field spectra and not from Hyperion. We will make this point explicit in the new version of the manuscript.*

**5. Page 6, Lines 28–29: "The only relevant discrepancy . . . where ASD spectra remain almost flat." Are there any possible explanations for this discrepancy at short wavelengths?**

*We made some hypothesis in line 5-9 (page 9). The observed discrepancy could be due to the presence of contaminated (non-pure) pixels of snow and ice, as previously observed by Negi et al. (2013). Otherwise, it could be related to the presence of meltwater increasing the absorption of solar radiation during the melting season, as observed from airborne hyperspectral reflectance data in other glaciers of the European Alps (Naegeli et al., 2015).*

**6. Page 7, Section 3.2: The authors only presented the concentration of EC and OC in this section, which seems to lack of the descriptions on the linkage between EC/OC concentration and reduced reflectance. This may confuse the readers. It would be helpful if the authors could explicitly articulate the relationship between EC/OC content and albedo reduction, after the description of EC/OC concentrations in this section.**

*Unfortunately, this comparison is not possible at the moment. In cryoconite, EC/OC are mixed with a mineral fraction that also reduce the reflectance. In this section, we meant to present the EC/OC concentration data since they can represent an important contribution to the overall albedo reduction. Decoupling the effect of mineral and organic fraction in cryoconite is a very difficult task, and it is out of the scope of the paper. Furthermore, no EC concentration in cryoconite are present in the scientific literature till now. Studying the carbonaceous fraction of cryoconite is an important task in order to estimate the impact of anthropogenic and natural activity on glacier darkening. This is also valid for ice sheets margins, where the "bio-albedo" feedback of cryoconite material has been recently acknowledged (see Cook et al. 2017 The Cryosphere Discuss.).*

**7. Page 2, Lines 5–10: for the authors' information, a recent study (Lee et al., 2016) combined satellite measurements and model simulations to show the reduced snow albedo caused by impurities over the southern Tibetan Plateau, which could be cited here as a useful reference source.**

**Reference:**

**Lee, W. L., K. N. Liou, C. He, S.-C. Liang, Z. Liu, Q. Yue (2016): Impact of absorbing aerosol deposition on snow albedo reduction over the southern**

**Tibetan Plateau Based on Satellite Observations, Theor. Appl. Climatol., 1-10, 10.1007/s00704-016-1860-4**

*Thank you for the suggestion, we will add this paper to the bibliography.*

*Best regards*

*Biagio Di Mauro & co-authors*

---

## Referee Comment (RC2) · Anonymous Referee #2 · 26 Jun 2017

Dear all,

This paper aims to combine field and satellite reflectance measurements with laboratory analyses of ice and cryoconite samples in order to map various impurities over a Swiss glacier. This is a worthy research aim, and the authors have undoubtedly produced a valuable dataset that will be relevant for future method development in impurity mapping. The paper is generally clearly written, its purpose is well articulated and the subject matter is appropriate for The Cryosphere. Ultimately, I would be pleased to see a version of this paper published.

There are, however, some issues that the authors ought to address prior to publication:

[Figure]

1. More details are required regarding the measurement protocol used to obtain spectral reflectance. What was the viewing angle? How was the fibre optic levelled? What was the footprint size of each measurement? Were the sample surfaces flat? The maximum clean ice visible reflectance in Fig.2 exceeded 1.3 – does this indicate that an oblique viewing angle or sloping surface caused the measurement to be near the forward scattering peak? How does the measurement angle compare with that of the Hyperion satellite?

2. I have some questions regarding the Snow Darkening Index (SDI). This measure is a ratio of blue and green reflectance values where more positive SDI is interpreted as high impurity load and vice versa. However, wet cryoconite has a near-flat spectrum across the blue and green wavelengths and will therefore have a low or negative SDI despite being very dark. In this case, the SDI cannot reliably distinguish between very clean and very dirty snow/ice. This is illustrated in Fig 2. Similarly, in Figure 2B the SDI would be lower for the wet cryoconite than the dry cryoconite despite it being much darker. Wouldn't the index also change as the snow or ice grains evolve, even when impurity loading remains constant simply because grain evolution preferentially alters reflectance in red-NIR wavelengths?

3. I also wonder about the use of SDI as a measure of mineral dust loading, compared to total impurities measured using Iimp? Mineral dusts, organic carbon, living algae, black carbon and mineral dusts all depress reflectance in the visible wavelengths and would all have similar effects on the SDI. Perhaps I have misunderstood, but it seems that SDI and Iimp are only arbitrarily different metrics. Presumably the different wavelengths lead to the metric having different sensitivities, but is there a meaningful difference in what they represent physically?

4. Reflectance across (most of) the visible wavelengths was integrated to provide albedo. However, integrating only the visible wavelengths omits a significant fraction of the total solar radiation that is crucial for the surface energy balance with the effect of exaggerating the albedo lowering effect of impurities. Albedo is also hemispheric.

Since ice is strongly forward scattering, large errors can result from assuming nadir reflectance can be integrated over the entire hemisphere without anisotropy correction. Was this accounted for in the analysis? If so, how? If not, the albedo discussion needs to be removed or heavily caveated.

5. I also agree with Reviewer 1 that intrinsic albedo reducing processes could influence the interpretation of the presented spectral data.

6. The albedo/spectral reflectance of cryoconite on the laboratory is likely to be very different to cryoconite in nature, especially when contained within cryoconite holes. Not only are the hole floors and walls made of ice with certain optical properties, the cryoconite material is usually submerged beneath a layer of water. This introduces specular reflection from the water surface, multiple reflections between hole walls and hides the cryoconite from light arriving from off-nadir angles. For cryoconite out of holes, its albedo influence will vary greatly depending upon the optical properties of the ice beneath it and how wet it is. For these reasons, care should be taken when inferring cryoconite's enhanced albedo-lowering effect relative to moraine sediment (page 10).

Specific Comments

Page 3, line 29: In what window around midday? Did you use midday or solar noon?

Page 4, line 14: Shouldn't they be normalised for mass, not concentration?

Page 6 line 29: These are also the wavelengths where both the incoming solar irradiance and snow and ice albedo peak. Are you confident the discrepancy does not lead to significant error?

---

## Author Comment (AC2) · 1 Sep 2017

Authors responses are in *italic,* Reviewer's comments are in **bold.**

**Dear all,**

**This paper aims to combine field and satellite reflectance measurements with laboratory analyses of ice and cryoconite samples in order to map various impurities over a Swiss glacier. This is a worthy research aim, and the authors have undoubtedly produced a valuable dataset that will be relevant for future method development in impurity mapping. The paper is generally clearly written, its purpose is well articulated and the subject matter is appropriate for The Cryosphere. Ultimately, I would be pleased to see a version of this paper published. There are, however, some issues that the authors ought to address prior to publication.**

*Dear Reviewer #2,*

*Thank you for the positive evaluation of the manuscript. We have carefully considered each of the Reviewer's comments and suggestions. The Reviewer will find below the responses to the general and specific comments.*

**General comments:**

**1. More details are required regarding the measurement protocol used to obtain spectral reflectance. What was the viewing angle? How was the fibre optic levelled? What was the footprint size of each measurement? Were the sample surfaces flat? The maximum clean ice visible reflectance in Fig.2 exceeded 1.3 – does this indicate that an oblique viewing angle or sloping surface caused the measurement to be near the forward scattering peak? How does the measurement angle compare with that of the Hyperion satellite?.**

*1. We acquired spectral measurements of the glacier surface from nadir using a bare fiber optic with an angular field of view of 25°. The fiber optic was held using a fiber holder equipped with a level to ensure that the glacier surface was always measured from nadir. Measurements were collected at a distance of 80 cm from the ground corresponding to a footprint diameter of 35 cm. We tried to select flat areas for the reflectance measurements, but nevertheless the surface of the glacier was quite rugged, so possible uncertainties related to the forward scattering of snow may be present in the data. Reflectance higher than 1 in the visible wavelengths is often found in the literature, and can be a symptom of this issue (Painter & Dozier 2004, Schaepman-Strub et al. 2006). The look angle of the Hyperion tile (EO1H1930282015219110K5_SG1_01) was 23°, this could further explain some differences between field and satellite observations.*

*This information was added in Section 2.2, now it reads (pg3 ln33):*

*"A bare optical fiber with a field of view of 25° was used to collect data from nadir with respect to the surface. The fiber optic was held by a fiber holder equipped with a level. The fiber holder was always kept at a distance of 80 cm from the ground corresponding to a footprint diameter of 35 cm. As a measure of the ASD reflectance measurements uncertainty, we calculated the coefficient of variation averaged on the VIS-NIR wavelengths. In our study, the coefficient of variation spans from 1 to 10%."*

*The information on the Hyperion look angle was added in Section 2.4 (pg5 ln12):*

*"The look angle of Hyperion was 23° during the acquisition"*

*The discussion regarding the reflectance was added in Section 4, now it reads (pg9 ln17):*

*"From field spectroscopy, we were able to characterize different glacier components in the ablation zone only, while satellite data allowed to have an overview on the reflectance spatial variability at catchment scale. We tried to select flat areas for the reflectance measurements. However, the surface of the glacier was quite rugged, so possible uncertainties related to the forward scattering of snow may be present in the data. Reflectance higher than 1 in the visible wavelengths is often found in the literature (Painter and Dozier, 2004; Schaepman-Strub et al., 2006), and can be a symptom of this issue."*

*References:*

*Painter, T. H., & Dozier, J. (2004). Measurements of the hemispherical-directional reflectance of snow at fine spectral and angular resolution. Journal of Geophysical Research: Atmospheres, 109(D18).*

*Schaepman-Strub, G., Schaepman, M. E., Painter, T. H., Dangel, S., & Martonchik, J. V. (2006). Reflectance quantities in optical remote sensing—Definitions and case studies. Remote sensing of environment, 103(1), 27-42.*

**2. I have some questions regarding the Snow Darkening Index (SDI). This measure is a ratio of blue and green reflectance values where more positive SDI is interpreted as high impurity load and vice versa. However, wet cryoconite has a near-flat spectrum across the blue and green wavelengths and will therefore have a low or negative SDI despite being very dark. In this case, the SDI cannot reliably distinguish between very clean and very dirty snow/ice. This is illustrated in Fig 2. Similarly, in Figure 2B the SDI would be lower for the wet cryoconite than the dry cryoconite despite it being much darker. Wouldn't the index also change as the snow or ice grains evolve, even when impurity loading remains constant simply because grain evolution preferentially alters reflectance in red-NIR wavelengths?**

**3. I also wonder about the use of SDI as a measure of mineral dust loading, compared to total impurities measured using Iimp? Mineral dusts, organic carbon, living algae, black carbon and mineral dusts all depress reflectance in the visible wavelengths and would all have similar effects on the SDI. Perhaps I have misunderstood, but it seems that SDI and Iimp are only arbitrarily different metrics. Presumably the different wavelengths lead to the metric having different sensitivities, but is there a meaningful difference in what they represent physically?**

*We addressed point 2 and 3 together since they are both related to the sensitivity of SDI and the other indices to LAIs and snow grain size.*

*SDI was developed to link the concentration of mineral dust (MD) with the spectral reflectance of snow. This index was specifically built to exploit the wavelength-dependent effect of MD on snow reflectance (see Di Mauro et al. 2015). In the context of this paper, the interesting information brought by SDI is related to the resurfacing of Saharan dust layers in the accumulation zone of the glacier. In the ablation zone, the presence of different materials (fine debris, cryoconite sediment etc.) makes the interpretation of the spatial distribution of the index quite difficult. We discussed these aspects in Section 4 (pg11 ln2).*

*Although SDI ad $I_{imp}$ share a common band (at 550-580 nm), they emphasize different aspects of the impact of LAIs on snow and ice reflectance. For example, Black Carbon (BC) and Organic Carbon (OC) depress the reflectance of snow in a more homogeneous way, and their effect is negligible in the NIR and SWIR wavelengths. Instead, MD strongly decreases the reflectance at wavelengths shorter than 500 nm. We acknowledge that SDI may change also in response to grain growth (see Fig. 8 of Di Mauro et al. 2015). However, the effect of SGS is evident in the NIR range while changes in the grain size can only slightly affect the visible wavelengths involved in the SDI computation.*

*In the "Data and Methods", we added Section 2.5 "Radiative transfer modelling" to explain the different sensitivity of SDI, $I_{imp}$ and $\alpha_{VIS}$ to grain size and LAI concentrations:*

*"In order to assess the sensitivity of the SDI, $I_{imp}$ and $\alpha_{VIS}$ to snow grain size (SGS), BC and MD concentrations, we ran a set of simulations using the Snow, Ice, and Aerosol Radiation (SNICAR) model (Flanner et al. 2007). The model allows to simulate the snow hemispherical albedo spectra between 300 and 5000 nm with a resolution of 10 nm. The main variables included in the model are: snow grain size (μm), snow density (Kg/m³), snowpack thickness (m), surface spectral distribution, solar zenith angle (degrees), MD and BC concentration (respectively in ppm and ppb). We simulated snow reflectance varying the SGS from 100 to 600μm, the (uncoated) BC concentration from 0 to 1200ppb and the MD concentration from 0 to 300ppm (diameter 5.0-10.0 μm). Then we calculated the three indices and represented them as a function of MD/BC concentrations and SGS in a contour matrix plot."*

In the "Results", we added Section "3.1.3 Sensitivity of narrow- and broad-band indices to SGS and LAI concentrations" describing the results of the SNICAR simulations:

"In Figure 8, we present the contour plots obtained from the SNICAR simulations. Plots refer to the sensitivity of narrow- and broad-band indices to MD variations (upper panels) and to BC variations (lower panels). $I_{imp}$ is insensitive to SGS for both MD and BC variations. For low concentrations of BC/MD, also SDI is almost insensitive to SGS, but for high concentrations, a nonlinearity emerges. $\alpha_{VIS}$ results the most sensitive index to SGS. $I_{imp}$ and $\alpha_{VIS}$ are similarly affected by variations in MD and BC, while SDI is more sensitive to MD than BC."

[Figure]

Figure 8 Comparison between SDI, $I_{imp}$ and $\alpha_{VIS}$ obtained from SNICAR simulations. The indexes are represented as contour plots as a function of the concentration of Mineral Dust (MD, upper panels) and Black Carbon (BC, lower panels). MD concentrations are in ppm, BC concentrations are in ppb.

In the manuscript, we added some discussion on this aspect in Section 4 (pg10 ln21):

"From the results of the SNICAR simulations presented in Section 3.1.3 we can assess that $I_{imp}$ is a solid indicator of LAIs concentration. SDI instead is more related to the radiative impact of LAIs on snow, since it is also influenced by the increase of SGS. However, during summer season characterized by wet snow with large SGS, SDI is almost insensitive to changes in SGS, while its sensitivity to medium/low MD concentration is maximum. Furthermore, since SDI is a broad band RGB index, it can be easily estimated from digital RGB cameras both fixed (e.g. Webcam) and mounted on Unmanned Aerial Vehicles.

Although SDI and $I_{imp}$ share a common band (at 550-580nm), they emphasize different aspects of the impact of LAIs on snow and ice reflectance. $I_{imp}$

*is similarly affected by variations in MD/BC, while SDI is more sensitive to MD variations, in particular for large SGS."*

*We added also a comment on the influence of sun geometry on SDI and $I_{imp}$ spatial variability (pg10 ln29):*

*"In comparing SDI and $I_{imp}$ maps, it should be noted also that both indices are varying with the sun geometry, which is varying on the glacier due to local topography. For example, Dumont et al. (2014) computed $I_{imp}$ from MODIS diffuse albedo to analyse LAIs distribution over the Greenland Ice Sheet."*

*References:*

*Di Mauro, B., Fava, F., Ferrero, L., Garzonio, R., Baccolo, G., Delmonte, B., & Colombo, R. (2015). Mineral dust impact on snow radiative properties in the European Alps combining ground, UAV, and satellite observations. Journal of Geophysical Research: Atmospheres, 120(12), 6080-6097.*

*Flanner, M. G., Zender, C. S., Randerson, J. T., & Rasch, P. J. (2007). Present-day climate forcing and response from black carbon in snow. Journal of Geophysical Research: Atmospheres, 112(D11).*

**4. Reflectance across (most of) the visible wavelengths was integrated to provide albedo. However, integrating only the visible wavelengths omits a significant fraction of the total solar radiation that is crucial for the surface energy balance with the effect of exaggerating the albedo lowering effect of impurities. Albedo is also hemispheric. Since ice is strongly forward scattering, large errors can result from assuming nadir reflectance can be integrated over the entire hemisphere without anisotropy correction. Was this accounted for in the analysis? If so, how? If not, the albedo discussion needs to be removed or heavily caveated.**

*Thank you for this comment. In this paper, we were interested in estimating visible albedo ($\alpha_{VIS}$) for studying the impact of impurities and cryoconite on snow and ice reflectance. $\alpha_{VIS}$ was computed according to the wavelength limits used in Liang et al. (2001), where $\alpha_{VIS}$ is estimated from reflectance in visible wavelengths (0.4 – 0.7 µm).*

*We are aware that albedo is also hemispheric, but hemispherical albedo is difficult to determine from satellite sensors because measurements are performed with fixed solar and viewing angles. Unfortunately, we did not characterize the anisotropy of snow on the glacier, so we were not able to perform a proper estimate of hemispherical albedo (see Naegeli et al. 2015), but we use a numerical integral of HCRF in visible wavelengths (0.4 – 0.7 µm), namely $\alpha_{VIS}$, as an approximation of snow albedo. The paper is more focused on the impact of impurities on the spectral reflectance of snow and ice on the Morteratsch glacier, we made this clear in the Discussion section of the paper.*

*We point out these aspects in the methodology section of the new version of the manuscript, that now reads (pg6 ln8):*

*"In this study, we did not characterize the anisotropy of snow on the glacier, so we were not able to perform a proper estimate of hemispherical albedo (see Naegeli et al. 2015), but we used $\alpha_{VIS}$ computed as the numerical integral of HCRF in visible wavelengths (0.4 – 0.7 µm) (Liang 2001), as an approximation of snow albedo."*

*References:*

*Liang, S. (2001). Narrowband to broadband conversions of land surface albedo I: Algorithms. Remote sensing of environment, 76(2), 213-238.*

*Naegeli, K., Damm, A., Huss, M., Schaepman, M., & Hoelzle, M. (2015). Imaging spectroscopy to assess the composition of ice surface materials and their impact on glacier mass balance. Remote Sensing of Environment, 168, 388-402.*

**5. I also agree with Reviewer 1 that intrinsic albedo reducing processes could influence the interpretation of the presented spectral data.**

*Thank you for this comment. We acknowledge that both internal and external factors impact snow and ice spectral reflectance. In particular, internal factors may play an important role in decreasing the reflectance of ice and snow during long and hot summers at mid-latitudes. We added a brief discussion on these aspects and we included the papers suggested by the Reviewer #1 in the bibliography. This point is also addressed in the new "Data and Methods" and "Results" sections based on SNICAR simulations. Furthermore, we added the following sentences (pg10 ln34):*

*"This decrease in albedo can be explained by both an increase of LAI content and/or variations of the snow/ice grain properties. However, the interpretation of the effects of such external and internal snow characteristics on $\alpha_{VIS}$ is not straightforward. For example, Liou et al. (2014) showed that snow grain shape and impurity snow mixing structures can significantly influence the effects of LAIs on snow albedo. Furthermore, snow grain packing also plays a critical role in affecting albedo of both clean and dirty snow (He et al., 2017)"*

**6. The albedo/spectral reflectance of cryoconite on the laboratory is likely to be very different to cryoconite in nature, especially when contained within cryoconite holes. Not only are the hole floors and walls made of ice with certain optical properties, the cryoconite material is usually submerged beneath a layer of water. This introduces specular reflection from the water surface, multiple reflections between hole walls and hides the cryoconite from light arriving from off-nadir angles. For cryoconite out of holes, its albedo influence will vary greatly depending upon the optical properties of the ice beneath it and how wet it is. For these reasons, care should be taken when inferring cryoconite's enhanced albedo-lowering effect relative to moraine sediment (page 10).**

*In this part of the paper we were interested in the optical properties of the materials that constitute cryoconite sediments. This characterization is fundamental for future developments of glacier modelling that take into account the impact of these materials that have not been extensively studied till now. Besides the large literature regarding the biological constituents of cryoconite, little attention has been paid to the geochemical and mineralogical properties of cryoconite (see Tedesco et al. 2012; Baccolo et al. 2017). Nevertheless, the bulk constituent of cryoconite is often composed by inorganic materials that, coupled with organic materials, trigger their development, and determine the radiative impact on glacier ablation. Nevertheless, we remark that field spectroscopy data acquired on the Morteratsch glacier were collected on surface cryoconite, not cryoconite holes. Samples from cryoconite holes were analysed only in laboratory with the ASD spectrometer and the Multi-Wavelength Absorbance Analyzer (MWAA). We made this clear in the revised paper.*

*In the "Discussion" section, we added the following sentence (pg9 ln37):*

*"However, it should be noted that wet cryoconite reflectance is expected to vary as a function of the optical properties of the ice beneath and its wetness, thus the effect of cryoconite presence on glacier albedo is not easily predictable"*

*Regarding the comparison between cryoconite and moraine sediments, results from ASD and MWAA analyses showed that the two materials show substantial differences. In particular, the fact that cryoconite absorbs more radiation with respect to moraine sediments implies that the organic material contained in the cryoconite strongly alters its optical properties.*

*References:*

*Baccolo, G., Di Mauro, B., Massabò, D., Clemenza, M., Nastasi, M., Delmonte, B., Prata, M., Prati, P., Previtali, E. and Maggi, V.: Cryoconite as a temporary sink for anthropogenic species stored in glaciers, Sci. Rep., 7(1), 9623, doi:10.1038/s41598-017-10220-5, 2017.*

*Tedesco, M., Foreman, C. M., Anton, J., Steiner, N., & Schwartzman, T. (2013). Comparative analysis of morphological, mineralogical and spectral properties of cryoconite in Jakobshavn Isbrae, Greenland, and Canada Glacier, Antarctica. Annals of Glaciology, 54(63), 147-157.*

**Specific Comments**

**Page 3, line 29: In what window around midday? Did you use midday or solar noon?**

*We collected the measurements between 12.00 and 13.00 (local time). We added this information in the revised paper (pg3 ln31).*

**Page 4, line 14: Shouldn't they be normalised for mass, not concentration?**

*Yes, you are right. We normalised for the total mass. We modified the sentence (pg4 ln25).*

**Page 6 line 29: These are also the wavelengths where both the incoming solar irradiance and snow and ice albedo peak. Are you confident the discrepancy does not lead to significant error?**

*During the field measurements, each spectral acquisition was the mean of 15 spectra. In this way, we are confident to minimize the source of errors in the data. The discrepancy between ASD and Hyperion data in wavelengths lower than 500nm can be probably due to the spatial averaging (30m) of Hyperion, or to other issues discussed in pg10 ln6.*

*Best regards*

*Biagio Di Mauro & co-authors*

---

## Author Response (AR1)

Authors responses are in *italic,* Reviewer's comments are in **bold.**

**The authors used field campaigns and satellite hyperspectral data to investigate the effects of impurities and cryoconite on spectral reflectance of snow and ice. They also conducted lab measurements of optical properties of ice and cryoconite samples, which is related to the impurity content in snow/ice. This study provides a good method to characterize the impact of impurities on snow/ice spectral reflectance by combining field, lab, and satellite measurements, which have an important implication for future study. Before this manuscript can be considered for publication, I have a few comments for the authors to address.**

*Dear Reviewer #1,*

*Thank you for the positive evaluation of the manuscript. We have carefully considered each of the Reviewer's comments and suggestions. The Reviewer will find below the responses to the specific comments.*

**General comments:**

**1. In the methodology section, the authors provided a detailed description of laboratory, field, and satellite measurement processes, which, however, lacks necessary discussions on the uncertainties associated with these measurements. I suggest that the authors add some discussions on this aspect.**

*Thank you for this comment. All our measurements feature different uncertainties. Hereafter, the uncertainty related to our measurements is discussed. We added this information in the Methodology section of the manuscript.*

*For the gravimetric determination of cryoconite concentration we estimated an error equal to 4% by repeating the measurement 5 times. Regarding EC/OC determination, uncertainty values are estimated from the software of the instrument, and are generally equal to 8-10%; referencing from the manual: "Calculated errors are based on long-term historical data for replicates and instrumental blanks. Over the course of hundreds of replicate runs, the relative standard deviation is typically 5%". For MWAA measurements, the uncertainty is about 10%, and it is given by the squared sum of the uncertainty related to the surface variability of the sample (~5%) and the uncertainty of the optical measurement (calculated as 3 times the variability of the blanks, and equal to 8%).*

*Regarding ASD data, measurement error is reduced by internally averaging 15 scans for each acquired spectrum, all corrected for the instrument dark current (See Pag. 3 lines 27-28). Field spectra were acquired at midday in clear sky conditions, so that the uncertainty related to variability of the incoming radiation should be in principle minimized. Furthermore, three replicates were collected for each sample. As a measure of the ASD reflectance measurements uncertainty, we calculated the coefficient of variation (ratio between the standard deviation and the mean, x 100) averaged on the VIS-NIR wavelengths. In our study, the coefficient of variation spans from 1 to 10%.*

*Hyperion measurements can be affected by different sources of uncertainties. Measurement uncertainty can be related to the sensor characteristics, as for example the signal-to-noise ratio (SNR) that varies from 150:1 (for 400-1000 nm) to 60:1 (for 1000-2000 nm). Other possible uncertainties may come from the atmospheric correction. In this study, we evaluated the reliability of the Hyperion reflectance through a comparison with Landsat and ASD reflectance, and we obtained respectively an RMSE of 0.015 and 0.03. We added this information in the new version of the manuscript, see below specific reference.*

*(pg3 ln36):*

*"As a measure of the ASD reflectance measurements uncertainty, we calculated the coefficient of variation averaged on the VIS-NIR wavelengths. In our study, the coefficient of variation spans from 1 to 10%."*

*(pg4 ln6):*

*"For the gravimetric determination of cryoconite concentration we estimated an error equal to 4% by repeating 5 times the measurement"*

*(pg4 ln25):*

*"For MWAA measurements, the uncertainty is about 10%, and it is given by the squared sum of the uncertainty related to the surface variability of the sample (~5%) and the uncertainty of the optical measurement (calculated as 3 times the variability of the blanks, and equal to 8%)."*

*(pg4 ln30):*

*"According to the instrument manual, an uncertainty of 8-10% is associated to the retrieved EC and OC concentrations."*

*(pg5 ln8):*

*"Hyperion reflectance retrievals have been validated several times with independent measurements, also with airborne sensor such as AVIRIS (Kruse et al., 2003).*

*(pg10 ln5)*

*"In this study, we evaluated the reliability of the Hyperion reflectance through a comparison with Landsat and ASD reflectance, and we obtained respectively an RMSE of 0.015 and 0.03."*

*Reference added:*

*Kruse, F. A., Boardman, J. W., & Huntington, J. F. (2003). Comparison of airborne hyperspectral data and EO-1 Hyperion for mineral mapping. IEEE Transactions on Geoscience and Remote Sensing, 41(6), 1388–1400.*

**2. The authors used the characteristic spectral reflectance of clean and dirty snow/ice to infer the effect of impurities in snow/ice. However, both external (e.g., impurity content) and internal (e.g., snow/ice grain properties) factors can affect the spectral reflectance. For example, Liou et al. (2014) showed that snow grain shape and impurity snow mixing structures can significantly influence the effects of impurities on snow albedo. He et al. (2017) further found that snow grain packing also plays a critical role in affecting albedos of both clean and dirty snow. Therefore, such internal factors could potentially affect the interpretation of the spectral observations presented by the authors. It would be informative and useful if the authors could include these recent studies and add some discussions on this issue.**

**Reference:**

**He, C., Y. Takano, and K. N. Liou (2017), Close packing effects on clean and dirty snow albedo and associated climatic implications, *Geophys. Res. Lett.*, 44, doi:10.1002/2017GL072916.**

Liou, K. N., Y. Takano, C. He, P. Yang, L. R. Leung, Y. Gu, and W. L. Lee (2014): Stochastic parameterization for light absorption by internally mixed BC/dust in snow grains for application to climate models, *J. Geophys. Res.-Atmos.*, 119, doi:10.1002/2014JD021665

*We acknowledge that both internal and external factors impact snow and ice spectral reflectance. In particular, internal factors may play an important role in decreasing the reflectance of ice and snow during long and hot summers at mid-latitudes. We added a brief discussion on these aspects and we included the suggested papers in the bibliography.*

*We added (pg10 ln34):*

*"This decrease in albedo can be explained by both an increase of LAI content and/or variations of the snow/ice grain properties. However, the interpretation of the effects of such external and internal snow characteristics on Avis is not straightforward. For example, Liou et al. (2014) showed that snow grain shape and impurity snow mixing structures can significantly influence the effects of LAIs on snow albedo. Furthermore snow grain packing also plays a critical role in affecting albedo of both clean and dirty snow (He et al. 2017)."*

*Furthermore, as requested by Reviewer #2, we added an in-depth analysis of the effects of mineral dust, black carbon and snow grain size on snow optical indices based on SNICAR simulations. The manuscript was modified adding the following paragraphs:*

*Methods (pg6 ln14)*

*2.5 Radiative transfer modelling*

*Results (pg8 ln18)*

*3.1.3 Sensitivity of narrow- and broad-band indices to SGS and LAI concentrations*

**Specific comments:**

**1. Page 3, Line 29: "The spectra were all obtained around midday under clear-sky conditions." Are there any specific reasons or advantages to obtain spectra in midday with clear sky?**

*Atmospheric disturbance is an important source of error in field spectroscopy. Incoming radiation in field environment is strongly anisotropic and it is a combination of direct/diffuse sunlight scattered from the sky and adjacent objects. These scattering events produce wavelength-dependent effects. As a consequence, HCRFs measured in the field can be affected by uncertainties introduced by variations in the irradiation environment, not related to the properties of the investigated surface (see Milton et al. 2009). For this reason, we measured field spectra in clear sky conditions in order to minimize the uncertainty related to the direct/diffuse ratio*

*during the field measurements. The choice of collecting measurements around midday is motivated by the fact that snow and ice have a strong directional effect (see for example Painter & Dozier 2004), and measuring with the Sun at nadir should minimize this source of error.*

*We addressed this aspect (pg3 ln31):*

*"HCRF spectra were all obtained between 12.00 and 13.00 (local time) under clear-sky conditions in order to minimize the uncertainty related to variations in the radiation environment (e.g., changes in the direct/diffuse ratio) during the field measurements"*

*References:*

*Milton, E. J., Schaepman, M. E., Anderson, K., Kneubühler, M., & Fox, N. (2009). Progress in field spectroscopy. Remote Sensing of Environment, 113, S92-S109.*

*Painter, T. H., & Dozier, J. (2004). Measurements of the hemispherical-directional reflectance of snow at fine spectral and angular resolution. Journal of Geophysical Research: Atmospheres, 109(D18).*

**2. Page 4, Line 4: "solid cryoconite was successively dried at 60∘C for 4 hours". Would this drying process remove some of the organics with relatively high volatility?**

*Even if we kept a relatively low drying temperature to avoid sample modifications, we cannot definitely exclude that some compounds with very high volatility may have been removed with the drying process. In any case, using the Sunset system we observed that organics do not volatilize at temperatures lower than 100°C. With this in mind, if we lost some organics with the drying process, they should be compounds that are in the gas phase at ambient temperature, and that usually constitute a minimum fraction with respect to the total OC.*

*We added (pg4 ln13):*

*"Even if we kept a relatively low drying temperature to avoid sample modifications, we cannot definitely exclude that some compounds with very high volatility may have been removed with the drying process. However, organics do not volatilize at temperatures lower than 100°C, thus, if we lost some organics with the drying process, they should be compounds that are in the gas phase at ambient temperature, and that usually constitute a minimum fraction with respect to the total OC."*

**3. Page 5, Lines 8–12: What is the percentage of total data points used for SVM training and testing set, respectively?**

*For the two main classes of interest (snow and bare ice), the ratio between training and test set pixels is ~ 10%. We included this information in the paper. The sentence now reads (pg5 ln27):*

*"For the two main classes of interest (snow and bare ice), the ratio between training and test set pixels was ~ 10%."*

**4. Page 5, Line 30: The indices (narrow- and broad-band) were compared to the impurity concentrations. The indices derived from the Hyperion spectra have a spatial resolution of 30 meters, while the impurity concentration is from point measurement. This is not an apple-to-apple comparison, which may introduce uncertainty. Could the authors discuss this issue?**

*For this comparison, indices were calculated from the ASD field spectra and not from Hyperion. We made this point explicit in the new version of the manuscript (pg6 ln12):*

*"These indices calculated from the ASD reflectance spectra were then compared to the concentration of impurities measured in clean and dirty ice"*

**5. Page 6, Lines 28–29: "The only relevant discrepancy . . . where ASD spectra remain almost flat." Are there any possible explanations for this discrepancy at short wavelengths?**

*We made some hypotheses at pg10 ln6-11. The observed discrepancy could be due to the presence of contaminated (non-pure) pixels of snow and ice, as previously observed by Negi et al. (2013). Otherwise, it could be related to the presence of meltwater increasing the absorption of solar radiation during the melting season, as observed from airborne hyperspectral reflectance data in other glaciers of the European Alps (Naegeli et al., 2015).*

**6. Page 7, Section 3.2: The authors only presented the concentration of EC and OC in this section, which seems to lack of the descriptions on the linkage between EC/OC concentration and reduced reflectance. This may confuse the readers. It would be helpful if the authors could explicitly articulate the relationship between EC/OC content and albedo reduction, after the description of EC/OC concentrations in this section.**

*Unfortunately, this comparison is not possible at the moment. In cryoconite, EC/OC are mixed with a mineral fraction that also reduces the reflectance. In this section, we meant to present the EC/OC concentration data since they can represent an important contribution to the overall albedo reduction. Decoupling the effect of mineral and organic fraction in cryoconite is a very difficult task, and it is out of the scope of the paper. Furthermore, no EC concentrations in cryoconite are present in the scientific literature till now. Studying the carbonaceous fraction of cryoconite is an important task in order to estimate the impact of*

*anthropogenic and natural activity on glacier darkening. This is also valid for ice sheets margins, where the "bio-albedo" feedback of cryoconite material has been recently acknowledged (see Cook et al. 2017 The Cryosphere Discuss.).*

**7. Page 2, Lines 5-10: for the authors' information, a recent study (Lee et al., 2016) combined satellite measurements and model simulations to show the reduced snow albedo caused by impurities over the southern Tibetan Plateau, which could be cited here as a useful reference source.**

**Reference:**

**Lee, W. L., K. N. Liou, C. He, S.-C. Liang, Z. Liu, Q. Yue (2016): Impact of absorbing aerosol deposition on snow albedo reduction over the southern Tibetan Plateau Based on Satellite Observations, Theor. Appl. Climatol., 1-10, 10.1007/s00704-016-1860-4**

*Thank you for the suggestion, we added a reference to this paper (pg2 ln8).*

*Best regards*

*Biagio Di Mauro & co-authors*
* * *
**Reviewer #2**

Response to Interactive comment on "Impact of impurities and cryoconite on the optical properties of the Morteratsch glacier (Swiss Alps)" by Biagio Di Mauro et al.

Anonymous Referee #2

Authors responses are in *italic,* Reviewer's comments are in **bold***.*

**Dear all,**

**This paper aims to combine field and satellite reflectance measurements with laboratory analyses of ice and cryoconite samples in order to map various impurities over a Swiss glacier. This is a worthy research aim, and the authors have undoubtedly produced a valuable dataset that will be relevant for future method development in impurity mapping. The paper is generally clearly written, its purpose is well articulated and the subject matter is appropriate for The Cryosphere. Ultimately, I would be pleased to see a version of this paper published. There are, however, some issues that the authors ought to address prior to publication.**

*Dear Reviewer #2,*

*Thank you for the positive evaluation of the manuscript. We have carefully considered each of the Reviewer's comments and suggestions. The Reviewer will find below the responses to the general and specific comments.*

**General comments:**

**1. More details are required regarding the measurement protocol used to obtain spectral reflectance. What was the viewing angle? How was the fibre optic levelled? What was the footprint size of each measurement? Were the sample surfaces flat? The maximum clean ice visible reflectance in Fig.2 exceeded 1.3 – does this indicate that an oblique viewing angle or sloping surface caused the measurement to be near the forward scattering peak? How does the measurement angle compare with that of the Hyperion satellite?.**

*1. We acquired spectral measurements of the glacier surface from nadir using a bare fiber optic with an angular field of view of 25°. The fiber optic was held using a fiber holder equipped with a level to ensure that the glacier surface was always measured from nadir. Measurements were collected at a distance of 80 cm from the ground corresponding to a footprint diameter of 35 cm. We tried to select flat areas for the reflectance measurements, but nevertheless the surface of the glacier was quite rugged, so possible uncertainties related to the forward scattering of snow may be present in the data. Reflectance higher than 1 in the visible wavelengths is often found in the literature, and can be a symptom of this issue (Painter & Dozier 2004, Schaepman-Strub et al. 2006). The look angle of the Hyperion tile (EO1H1930282015219110K5_SG1_01) was 23°, this could further explain some differences between field and satellite observations.*

*This information was added in Section 2.2, now it reads (pg3 ln33):*

*"A bare optical fiber with a field of view of 25° was used to collect data from nadir with respect to the surface. The fiber optic was held by a fiber holder equipped with a level. The fiber holder was always kept at a distance of 80 cm from the ground corresponding to a footprint diameter of 35 cm. As a measure of the ASD reflectance measurements uncertainty, we calculated the coefficient of variation averaged on the VIS-NIR wavelengths. In our study, the coefficient of variation spans from 1 to 10%."*

*The information on the Hyperion look angle was added in Section 2.4 (pg5 ln12):*

*"The look angle of Hyperion was 23° during the acquisition"*

*The discussion regarding the reflectance was added in Section 4, now it reads (pg9 ln17):*

*"From field spectroscopy, we were able to characterize different glacier components in the ablation zone only, while satellite data allowed to have an overview on the reflectance spatial variability at catchment scale. We tried to select flat areas for the reflectance measurements. However, the surface of the glacier was quite rugged, so possible uncertainties related to the forward scattering of snow may be present in the data. Reflectance higher than 1 in the visible wavelengths is often found in the literature (Painter and Dozier, 2004; Schaepman-Strub et al., 2006), and can be a symptom of this issue."*

In the "Results", we added Section "3.1.3  Sensitivity of narrow- and broad-band indices to SGS and LAI concentrations" describing the results of the SNICAR simulations:

"In Figure 8, we present the contour plots obtained from the SNICAR simulations. Plots refer to the sensitivity of narrow- and broad-band indices to MD variations (upper panels) and to BC variations (lower panels). $I_{imp}$ is insensitive to SGS for both MD and BC variations. For low concentrations of BC/MD, also SDI is almost

*insensitive to SGS, but for high concentrations, a nonlinearity emerges. $\alpha_{VIS}$ results the most sensitive index to SGS. $I_{imp}$ and $\alpha_{VIS}$ are similarly affected by variations in MD and BC, while SDI is more sensitive to MD than BC."*

[Figure]

*Figure 8 Comparison between SDI, $I_{imp}$ and $\alpha_{VIS}$ obtained from SNICAR simulations. The indexes are represented as contour plots as a function of the concentration of Mineral Dust (MD, upper panels) and Black Carbon (BC, lower panels). MD concentrations are in ppm, BC concentrations are in ppb.*

In the manuscript, we added some discussion on this aspect in Section 4 (pg10 ln21):

*"From the results of the SNICAR simulations presented in Section 3.1.3 we can assess that $I_{imp}$ is a solid indicator of LAIs concentration. SDI instead is more related to the radiative impact of LAIs on snow, since it is also influenced by the increase of SGS. However, during summer season characterized by wet snow with large SGS, SDI is almost insensitive to changes in SGS, while its sensitivity to medium/low MD concentration is maximum. Furthermore, since SDI is a broad band RGB index, it can be easily estimated from digital RGB cameras both fixed (e.g. Webcam) and mounted on Unmanned Aerial Vehicles.*

*Although SDI and $I_{imp}$ share a common band (at 550–580nm), they emphasize different aspects of the impact of LAIs on snow and ice reflectance. $I_{imp}$ is similarly affected by variations in MD/BC, while SDI is more sensitive to MD variations, in particular for large SGS."*

We added also a comment on the influence of sun geometry on SDI and $I_{imp}$ spatial variability (pg10 ln29):

*"In comparing SDI and $I_{imp}$ maps, it should be noted also that both indices are varying with the sun geometry, which is varying on the glacier due to local topography. For example, Dumont et al. (2014)*

computed $I_{imp}$ from MODIS diffuse albedo to analyse LAIs distribution over the Greenland Ice Sheet."

*In this part of the paper we were interested in the optical properties of the materials that constitute cryoconite sediments. This characterization is fundamental for future developments of glacier modelling that take into account the impact of these materials that have not been extensively studied till now. Besides the large literature regarding the biological constituents of cryoconite, little attention has been paid to the geochemical and mineralogical properties of cryoconite (see Tedesco et al. 2012; Baccolo et al. 2017). Nevertheless, the bulk constituent of cryoconite is often composed by inorganic materials that, coupled with organic materials, trigger their development, and determine the radiative impact on glacier ablation. Nevertheless, we remark that field spectroscopy data acquired on the Morteratsch glacier were collected on surface cryoconite, not cryoconite holes. Samples from cryoconite holes were analysed only in laboratory with the ASD spectrometer and the Multi-Wavelength Absorbance Analyzer (MWAA). We made this clear in the revised paper.*

*In the "Discussion" section, we added the following sentence (pg9 ln37):*

*"However, it should be noted that wet cryoconite reflectance is expected to vary as a function of the optical properties of the ice beneath and its wetness, thus the effect of cryoconite presence on glacier albedo is not easily predictable"*

*Regarding the comparison between cryoconite and moraine sediments, results from ASD and MWAA analyses showed that the two materials show substantial differences. In particular, the fact that cryoconite absorbs more radiation with respect to moraine sediments implies that the organic material contained in the cryoconite strongly alters its optical properties.*

**Specific Comments**

**Page 3, line 29: In what window around midday? Did you use midday or solar noon?**

*We collected the measurements between 12.00 and 13.00 (local time). We added this information in the revised paper (pg3 ln31).*

**Page 4, line 14: Shouldn't they be normalised for mass, not concentration?**

*Yes, you are right. We normalised for the total mass. We modified the sentence (pg4 ln25).*

**Page 6 line 29: These are also the wavelengths where both the incoming solar irradiance and snow and ice albedo peak. Are you confident the discrepancy does not lead to significant error?**

*During the field measurements, each spectral acquisition was the mean of 15 spectra. In this way, we are confident to minimize the source of errors in the data. The discrepancy between ASD and Hyperion data in wavelengths lower than 500nm can be probably due to the spatial averaging (30m) of Hyperion, or to other issues discussed in pg10 ln6.*

*Best regards*

*Biagio Di Mauro & co-authors*

[revised manuscript text omitted]

90%.**

---

## Referee Report (RR1)

**Review Report: tc-2017-66: Impact of impurities and cryoconite on the optical properties of the Morteratsch glacier (Swiss Alps), Biagio Di Mauro, Giovanni Baccolo, Roberto Garzonio, Claudia Giardino, Dario Massabò, Andrea Piazzalunga, Micol Rossini, and Roberto Colomb**

Dear Authors,

Thank you for your thorough response. The additional materials presented were very interesting and I think the revised manuscript will make a valuable addition to the literature. I am pleased to recommend the amended version for publication in The Cryosphere.

---

## Author Response (AR2)

Dear Editor,

thank you for the positive evaluation of the reviewed version of our paper. Here you find a point-by-point reply to your comments. Authors responses are in *italic*, Editor's comments are in **bold**.

**Page 6 line 10: If I understand correctly, while doing this calculation you are assuming that snow/ice is lambertian? If so, it might worth being explicit about it in the paper.**

*Yes, we are assuming snow and ice as Lambertian. We made it clear in the manuscript (ln 10 pg 6), that now reads:*

*"This calculation was performed assuming a Lambertian behaviour of snow and ice reflectance"*

**Section 2.5: The simulations have only be done for diffuse illumination? If so, it must also be stated in the paper. How would the results change for a different illumination?**

*We apologize for not being clear in the manuscript. We ran the SNICAR simulations using direct incident radiation (sun zenith angle = 50°). We now added this information to the revised paper. We used direct illumination in order to produce simulations more comparable to satellite measurements. The solar zenith angle was 50° during Hyperion acquisition. The impact of different illumination conditions on narrow- and broad-band indices is beyond the main objectives of the paper, so we prefer not to include this discussion to the revised paper. In the revised paper (Section 2), we added the information concerning the illumination conditions.*

*Ln 12, pg 5*

*" [..] and the solar zenith angle (SZA) was equal to 50°."*

*Ln 24, pg 6*

*"SNICAR simulations were run with a direct incident radiation (SZA = 50°) in order to match the Hyperion acquisition"*

**Page 9 – lines 16-21: "However …. related to the forward scattering.." I don't understand this sentence, maybe it needs to be reformulated. Reflectance higher than 1 can be also related to slope, sensor tilt, or inappropriate BRDF correction.**

*We agree with the Editor that the sentence was not clear. We reworded the sentence as follows (ln 25 pg 9):*

[revised manuscript text omitted]